# Epidemiology and transmission dynamics of multidrug-resistant organisms in nursing homes within the United States

Lona Mody [1,2] ✉, Kristen E. Gibson [1], Marco Cassone[1], Ganga Vijayasiri[1], Tasmine Clement[3], Evan Snitkin[1], Sanjay Saint[1,4], Sarah L. Krein[1,4], Mary R. Janevic[5], Jessica Thiel[2], Jennifer Ridenour[2], Alexandria Nguyen[6], Oteshia Hicks[6], Taissa A. Bej[6], Nadim G. El Chakhtoura[6], Lillian Min[1,2], Andrzej Galecki[1,7], Todd Greene[1,2], Mary-Claire Roghmann[8,9], Laxmi Chigurupati[10], Federico Perez[6] & Robin L. P. Jump[11,12]

Nursing home (NH) residents in the United States routinely attend interactive visits for services such as therapy or dialysis, creating opportunities for pathogen transmission. A paucity of studies exist which delineate spread of pathogens beyond residents' in-room environment. In this prospective cohort study, we recruited 197 newly-admitted residents across three Veterans Affairs NHs to characterize multidrug-resistant organism (MDRO) prevalence, acquisition, and transmission. Participant hands, nares, groin, and seven environmental surfaces were swabbed during 758 regularly scheduled in-room visits; participant hands, healthcare personnel hands, and equipment were swabbed during 345 unscheduled interactive visits. We demonstrate that baseline MDRO colonization and new acquisition is common, and one in six interactive visits result in MDRO transmission. Whole genome sequencing on a subset of participants enabled us to identify sources of transmission where it was unknown using microbiologic methods alone. Our results illustrate MDRO transmission pathways and highlight the need for innovative, multidisciplinary interventions.

Healthcare is provided in a variety of settings, including hospitals, outpatient clinics, skilled nursing facilities, rehabilitation centers, dialysis units, inpatient hospice centers, and other long-term care settings. Care is increasingly provided in post-acute care settings such as skilled nursing facilities. These facilities care for individuals who need rehabilitation, prolonged intravenous antibiotics, wound care, radiation therapy, end-of-life care, and other support. An explicit goal of these settings is to help residents regain function and be discharged to their homes[1–3].

[1]Department of Internal Medicine, University of Michigan Medical School, Ann Arbor, MI, USA. [2]Geriatric Research Education and Clinical Center (GRECC), Veterans' Affairs Ann Arbor Healthcare System, Ann Arbor, MI, USA. [3]Department of Computational Medicine and Bioinformatics, University of Michigan, Ann Arbor, MI, USA. [4]Department of Veterans' Affairs Center for Clinical Management Research, Ann Arbor, MI, USA. [5]Department of Health Behavior and Health Equity, University of Michigan School of Public Health, Ann Arbor, MI, USA. [6]Geriatric Research Education and Clinical Center (GRECC), VA Northeast Ohio Healthcare System, Cleveland, OH, USA. [7]Department of Biostatistics, University of Michigan School of Public Health, Ann Arbor, MI, USA. [8]Department of Epidemiology and Public Health, University of Maryland School of Medicine, Baltimore, MD, USA. [9]Geriatric Research Education and Clinical Center (GRECC), Veterans Affairs Maryland Health Care System, Baltimore, MD, USA. [10]John D. Dingell Department of Veterans Affairs Medical Center, Detroit, MI, USA. [11]Geriatric Research Education and Clinical Center (GRECC), VA Pittsburgh Healthcare System, Pittsburgh, PA, USA. [12]Division of Geriatric Medicine, Department of Medicine, University of Pittsburgh School of Medicine, Pittsburgh, PA, USA. ✉e-mail: lonamody@med.umich.edu

Post-acute and long-term care settings, hereby referred to as nursing homes (NHs), present a distinctive challenge to providing an environment that is conducive to effective infection prevention yet still maintaining a home-like atmosphere[4]. Unfortunately, infection rates and prevalence of multidrug-resistant organisms (MDROs) within NH settings are high, often surpassing rates in hospitals and intensive care units[5,6]. Despite the inherent vulnerability of NH residents and some early epidemiologic investigations[7–11], there remains a significant gap in understanding the clinical and genomic epidemiology of MDROs in this setting.

In the U.S., 132 Department of Veterans Affairs (VA) NHs, also known as community living centers, provide care for a frail, complex population, serving approximately 46,000 Veterans annually[12]. VA NHs are often co-located with VA medical centers and can provide a variety of services, such as rehabilitation therapy in a dedicated gym, dialysis, radiology, oncology treatments, dental care, and ophthalmology. These primarily out-of-room encounters (hereafter referred to as "interactive visits," as they require substantial hands-on contact and have opportunity for bidirectional transmission), allow us to investigate MDRO transmission dynamics involving residents, healthcare providers (HCPs), and the environment, particularly in settings beyond the resident's room and in shared environments.

Using surveillance cultures and genomic methods, the primary objectives of this study were to: (1) define the epidemiology and longitudinal changes in carriage of various MDROs; (2) assess the transmission of MDROs during a variety of interactions; and (3) integrate comprehensive genomic data to identify drivers of MDRO transmission across different healthcare settings. In this work we show that baseline MDRO prevalence (on participants and in the environment) is high, new acquisition rates (on participants and in the environment) are high, and transmission of MDROs during interactive visits outside the resident room is common, highlighting the urgent need for multimodal interventions to reduce MDRO burden and curb transmissions.

## Results
### Cohort description
We enrolled 200 (38.3%) of 522 potentially eligible participants recently admitted to one of the three participating NHs. The main reasons for non-enrollment were resident refusal ($n = 269$), and inability to obtain consent within three days of admission ($n = 61$). Three enrolled participants were removed from the analysis due to absence of HIPAA authorization, for a total sample size of 197 participants—98 from Facility A, 57 from Facility B, and 42 from Facility C (Supplementary Fig. 1). Of the 197 participants followed for 6480 total resident-days, we conducted 1103 study visits, of which 758 were scheduled, in-room visits (baseline or follow-up), assessing participants colonization and room contamination (mean 3.8/study participant; 2258 participant body site swabs; 5285 in-room environment swabs), and 345 were unscheduled, interactive visits with 135 participants (mean 2.6/participant; 919 participant hand swabs; 1674 surface swabs during interactive visits; 409 HCP hand swabs). Approximately 143 (41.4%) interactive visits were in the participant room or within the NH unit (including NH hallway and dining area), while 202 (58.6%) interactive visits occurred outside the participant's room, in a different department within the VA hospital.

Across all three cohorts, the average age of participants was 69.8 years (SD = 9.3); 96.4% were male; 64.2% were white, 32.6% Black or African American, and 3.2% another race/ethnicity (Table 1). The median length of stay in the study across all three cohorts was 28.0 days (interquartile range (IQR) = 13-41) and the median length of preadmission hospitalization was 8.0 days (IQR = 5-15). The study population had significant disability, devices, wounds, and comorbidities. Differences in participant race/ethnicity were evident across the three cohorts—Facility A was 81% white and 15% Black; Facility B was

59% white and 38% Black; Facility C was 33% white and 68% Black. Significant differences in baseline characteristics across the three cohorts were also detected for recent antibiotic use, wounds, PICC lines, length of preadmission hospitalization, length of stay in the study, median Katz score, and functional dependence in all activities of daily living (ADLs) assessed. Compared with Facilities A and B, Facility C had participants with fewer PICC lines, shorter preadmission hospitalization, and shorter length of stay in the study, but higher ADL scores corresponding to greater functional dependence (Table 1). Further univariate screening is presented at a facility level, stratified by MDRO colonization status at baseline and by new MDRO acquisition status in Supplementary Tables 1 and 2.

### MDRO participant colonization and environmental contamination at baseline
Over one-third of all study participants (72/197, 36.5%) were colonized with an MDRO (either methicillin-resistant *Staphylococcus aureus* (MRSA), vancomycin-resistant enterococci (VRE), or resistant gram-negative bacilli (RGNB)) at baseline (at any one of the three body sites surveyed: hand, nares, or groin): 8.6% with MRSA, 25.9% with VRE, and 13.7% with RGNB (Table 2). Baseline participant colonization at Facilities A, B, and C was: 7.1%, 12.3%, and 7.1% with MRSA; 27.6%, 28.0%, and 19.0% with VRE; 10.2%, 21.1%, and 11.9% with RGNB; and 35.7%, 43.9%, and 28.6% with any MDRO, respectively. Baseline environmental contamination for all study participants was similar, with 38.1% (75/197) of all participant rooms having at least one surface contaminated (of the seven surfaces sampled) with an MDRO (either MRSA, VRE, or RGNB), notably bedrail (31/197, 15.7%) or bed control contamination (31/197, 15.7%), followed by privacy curtain contamination (28/190, 14.7%). The most contaminated environmental sites at baseline at Facilities A, B, and C were, respectively: bedrail (15/98, 15.3%), privacy curtain (17/57, 29.8%), and bed control (8/42, 19.0%). The statistically significant differences in baseline colonization or contamination across the three facilities included curtain contamination with any MDRO ($p < 0.001$, Pearson's chi-square test), and VRE contamination at the nares ($p = 0.035$, Pearson's chi-square test), table top ($p = 0.033$, Pearson's chi-square test), or privacy curtain ($p = 0.014$, Pearson's chi-square test) (Table 2). Risk factors for baseline participant colonization with any MDRO included recent antibiotic use and having a prolonged hospitalization prior to NH admission (i.e., >14 days) (Table 3).

### MDRO colonization at baseline, discharge, and anytime
Fifteen study participants did not have any follow-up in-room visits completed after the baseline visit because they were discharged shortly after study enrollment; thus, these 15 participants are excluded from this analysis. Among 182 participants with at least one in-room follow-up visit, 36.8% (67/182) were colonized with an MDRO at baseline: 9.3% with MRSA, 25.8% with VRE, and 14.3% with RGNB; and 35.7% (65/182) were colonized with an MDRO at discharge (the last study visit): 11.0% with MRSA, 25.3% with VRE, and 9.9% with RGNB. Overall, the proportion of participants with MRSA, VRE, RGNB, or any MDRO colonization at any time during the study was almost twice that of baseline colonization (19.2% vs. 9.3% MRSA; 48.4% vs. 25.8% VRE; 30.2% vs. 14.3% RGNB; 65.4% vs. 36.8% any MDRO) (Fig. 1). Facility-level differences in patient colonization at baseline, at discharge, and at any time are shown as dots in Fig. 1; no significant differences in baseline, discharge, or any time colonization were detected across the three facilities. For further details on participant colonization across all in-room visits and across all interactive visits, including colonization of multiple body sites or organisms simultaneously, see Supplementary Figs. 2 (in-room visits) and 3 (interactive visits).

Overall, 23 of 115 (20.0%) participants not colonized with any MDRO at baseline later acquired and were discharged with at least one new MDRO. By contrast, 37.3% of participants who were colonized with

**Table 1 | Baseline characteristics of study participants, by facility**

| | Totalᵃ | All facilities (n = 197) | Facility A (n = 98) | Facility B (n = 57) | Facility C (n = 42) | P-value |
|---|---|---|---|---|---|---|
| **Demographics** | | | | | | |
| Age, years, mean (SD) | 197 | 69.8 (9.3) | 70.6 (9.3) | 68.0 (9.6) | 70.3 (8.5) | 0.230 |
| Male | 197 | 190 (96.4%) | 96 (98.0%) | 57 (100.0%) | 37 (88.1%) | 0.009 |
| Race/Ethnicity | 190 | | | | | 6.3e-8 |
| White | | 122 (64.2%) | 76 (80.9%) | 33 (58.9%) | 13 (32.5%) | |
| Black/African American | | 62 (32.6%) | 14 (14.9%) | 21 (37.5%) | 27 (67.5%) | |
| Other Race/Ethnicity | | 6 (3.2%) | 4 (4.3%) | 2 (3.6%) | 0 (0.0%) | |
| **Clinical Factors** | | | | | | |
| Recent Antibiotic Use | 197 | 137 (69.5%) | 77 (78.6%) | 33 (57.9%) | 27 (64.3%) | 0.019 |
| Wounds at Baseline | 197 | 68 (34.5%) | 32 (32.7%) | 27 (47.4%) | 9 (21.4%) | 0.024 |
| Any of UC/FT at Baseline | 197 | 32 (16.2%) | 14 (14.3%) | 12 (21.1%) | 6 (14.3%) | 0.506 |
| Uses Urinary Catheter | 197 | 27 (13.7%) | 11 (11.2%) | 10 (17.5%) | 6 (14.3%) | 0.540 |
| Uses Feeding Tube | 197 | 9 (4.6%) | 4 (4.1%) | 5 (8.8%) | 0 (0.0%) | 0.113 |
| Uses PICC Line | 197 | 40 (20.3%) | 24 (24.5%) | 14 (24.6%) | 2 (4.8%) | 0.010 |
| Charlson Score, median (IQR) | 197 | 3 (2-5) | 4 (2-5) | 3 (2-6) | 3 (1-5) | 0.122 |
| Length of Preadmission Hospitalization, days | 195 | | | | | 3.8e-4 |
| 0–3 days | | 32 (16.4%) | 21 (21.4%) | 1 (1.8%) | 10 (24.4%) | |
| 4–7 days | | 54 (27.7%) | 20 (20.4%) | 18 (32.1%) | 16 (39.0%) | |
| 8–14 days | | 60 (30.8%) | 30 (30.6%) | 24 (42.9%) | 6 (14.6%) | |
| >14 days | | 49 (25.1%) | 27 (27.6%) | 13 (23.2%) | 9 (22.0%) | |
| Recent hospitalization length, days, median (IQR) | 195 | 8 (5-15) | 10 (4-16) | 10.5 (6-14) | 6 (4-11) | 0.065 |
| Length of stay in study, days, median (IQR) | 197 | 28 (13-41) | 29 (15-40) | 35 (20-60) | 12.5 (6-30) | 3.0e-4 |
| Katz (ADL) Scoreᵇ, median (IQR) | 197 | 3.6 (2.1) | 3.6 (2.3) | 3.0 (2.0) | 4.5 (1.4) | 0.002 |
| Functional dependenceᶜ at baseline | | | | | | |
| Transferring | 197 | 152 (77.2%) | 73 (74.5%) | 40 (70.2%) | 39 (92.9%) | 0.012 |
| Bathing | 197 | 147 (74.6%) | 70 (71.4%) | 38 (66.7%) | 39 (92.9%) | 0.004 |
| Toileting | 197 | 139 (70.6%) | 67 (68.4%) | 34 (59.6%) | 38 (90.5%) | 0.002 |
| Dressing | 197 | 140 (71.1%) | 63 (64.3%) | 38 (66.7%) | 39 (92.9%) | 0.001 |
| Continence | 197 | 79 (40.1%) | 48 (49.0%) | 9 (15.8%) | 22 (52.4%) | 4.8e-5 |
| Feeding | 197 | 57 (28.9%) | 30 (30.6%) | 14 (24.6%) | 13 (31.0%) | 0.688 |
| Reason for Study Discharge | 197 | | | | | 0.460 |
| Discharged home | | 152 (77.2%) | 76 (77.6%) | 42 (73.7%) | 34 (81.0%) | |
| Discharged to VA or outside hospital | | 15 (7.6%) | 8 (8.2%) | 4 (7.0%) | 3 (7.1%) | |
| Discharged at resident/ family's request | | 4 (2.0%) | 3 (3.1%) | 1 (1.8%) | 0 (0.0%) | |
| Discharged at study staff's discretion | | 1 (0.5%) | 1 (1.0%) | 0 (0.0%) | 0 (0.0%) | |
| Discharged to hospice care | | 4 (2.0%) | 4 (4.1%) | 0 (0.0%) | 0 (0.0%) | |
| Other | | 3 (1.5%) | 0 (0.0%) | 2 (3.5%) | 1 (2.4%) | |
| Still resides at NH (finished 90 days of follow-up) | | 18 (9.1%) | 6 (6.1%) | 8 (14.0%) | 4 (9.5%) | |

Data are number of participants (column %), unless otherwise specified. *P*-values (two-sided) are based on ANOVA for continuous variables and Pearson's chi-square or Fisher's exact test for categorical variables. *P*-value (two-sided) for "Recent hospitalization length," is based on Kruskal-Wallis test.

*ADL* activities of daily living, *CRE* carbapenem-resistant Enterobacterales, *ESBL* extended spectrum beta-lactamase, *FT* feeding tube, *IQR* interquartile range, *MRSA* methicillin-resistant *Staphylococcus aureus*, *NH* nursing home, *PICC* peripherally inserted central catheter, *SD* standard deviation, *UC* urinary catheter, *VA* Veterans Affairs, *VRE* vancomycin-resistant enterococci.

ᵃDue to data missing on admission.

ᵇOverall Katz score ranges from 0-6, with 0 being independent and 6 being dependent in all six ADLs assessed.

ᶜDependence in function are defined as (1) transferring: needs help in moving from bed to chair or requires a complete transfer; (2) bathing: needs help with bathing more than one part of the body, getting in or out of the tub or shower, or requires total bathing assistance; (3) toileting: needs help transferring to the toilet, cleaning self or uses bedpan or commode;(4) dressing: needs help dressing self or needs to be completely dressed; (5) continence: is partially or totally incontinent of bowel or bladder; (6) feeding: needs partial or total help with feeding or requires parenteral feeding.

at least one MDRO at baseline were no longer detected with that MDRO at discharge (Fig. 2). At Facilities A, B, and C, respectively, 12 of 58 (20.7%), 7 of 31 (22.6%), and 4 of 26 (15.4%) participants not colonized with any MDRO at baseline later acquired and were discharged with a new MDRO. By contrast, at Facilities A, B, and C, respectively, 43.8%, 33.3%, and 27.3% of participants who were colonized with an MDRO at baseline were no longer detected at discharge (Supplementary Fig. 4). No statistically significant differences were seen across the three facilities. New acquisition and spontaneous loss of MDROs was a dynamic process during participants' stay in the NH.

## New Acquisition of MDROs

To estimate MDRO acquisition, we limited the analysis to the 185 participants with one or more in-room or interactive visits. Participants frequently acquired a new MDRO: MRSA (10.7% of 168 at-risk participants), VRE (29.7% of 138 at-risk), RGNB (18.4% of 158 at-risk), and any

**Table 2 | Participant colonization and environmental surface contamination at study baseline (during the first in-room visit)**

| | No. Participants with Baseline Colonization or Contamination / Total Participants (%) | | | | |
|---|---|---|---|---|---|
| **All MDROs** | **All Facilities** | **Facility A** | **Facility B** | **Facility C** | **P-value** |
| Nares | 19/197 (9.6%) | 7/98 (7.1%) | 9/57 (15.8%) | 3/42 (7.1%) | |
| Hand | 36/197 (18.3%) | 16/98 (16.3%) | 13/57 (22.8%) | 7/42 (16.7%) | |
| Groin | 56/195 (28.7%) | 26/98 (26.5%) | 21/56 (37.5%) | 9/41 (22.0%) | |
| Any Patient Body site[a] | 72/197 (36.5%) | 35/98 (35.7%) | 25/57 (43.9%) | 12/42 (28.6%) | 0.29 |
| Bed control | 31/197 (15.7%) | 13/98 (13.3%) | 10/57 (17.5%) | 8/42 (19.0%) | |
| Call button | 22/197 (11.2%) | 11/98 (11.2%) | 8/57 (14.0%) | 3/42 (7.1%) | |
| Table top | 23/197 (11.7%) | 9/98 (9.2%) | 11/57 (19.3%) | 3/42 (7.1%) | |
| TV remote/buttons | 20/197 (10.2%) | 8/98 (8.2%) | 8/57 (14.0%) | 4/42 (9.5%) | |
| Privacy curtain | 28/190 (14.7%) | 9/95 (9.5%) | 17/57 (29.8%) | 2/38 (5.3%) | |
| Bedrail | 31/197 (15.7%) | 15/98 (15.3%) | 11/57 (19.3%) | 5/42 (11.9%) | |
| Toilet seat | 23/197 (11.7%) | 11/98 (11.2%) | 10/57 (17.5%) | 2/42 (4.8%) | |
| Any Environmental site[b] | 75/197 (38.1%) | 35/98 (35.7%) | 28/57 (49.1%) | 12/42 (28.6%) | 0.091 |
| **MRSA** | | | | | |
| Nares | 9/197 (4.6%) | 4/98 (4.1%) | 2/57 (3.5%) | 3/42 (7.1%) | |
| Hand | 11/197 (5.6%) | 5/98 (5.1%) | 4/57 (7.0%) | 2/42 (4.8%) | |
| Groin | 5/195 (2.6%) | 0/98 (0) | 3/56 (5.4%) | 2/41 (4.9%) | |
| Any Patient Body site[a] | 17/197 (8.6%) | 7/98 (7.1%) | 7/57 (12.3%) | 3/42 (7.1%) | 0.51 |
| Bed control | 11/197 (5.6%) | 4/98 (4.1%) | 5/57 (8.8%) | 2/42 (4.8%) | |
| Call button | 8/197 (4.1%) | 2/98 (2.0%) | 4/57 (7.0%) | 2/42 (4.8%) | |
| Table top | 7/197 (3.6%) | 2/98 (2.0%) | 3/57 (5.3%) | 2/42 (4.8%) | |
| TV remote/buttons | 7/197 (3.6%) | 2/98 (2.0%) | 2/57 (3.5%) | 3/42 (7.1%) | |
| Privacy curtain | 8/190 (4.2%) | 2/95 (2.1%) | 5/57 (8.8%) | 1/38 (2.6%) | |
| Bedrail | 9/197 (4.6%) | 3/98 (3.1%) | 3/57 (5.3%) | 3/42 (7.1%) | |
| Toilet seat | 1/197 (0.5%) | 0/98 (0) | 0/57 (0) | 1/42 (2.4%) | |
| Any Environmental site[b] | 22/197 (11.2%) | 8/98 (8.2%) | 9/57 (15.8%) | 5/42 (11.9%) | 0.34 |
| **VRE** | | | | | |
| Nares | 7/197 (3.6%) | 2/98 (2.0%) | 5/57 (8.8%) | 0/42 (0) | |
| Hand | 28/197 (14.2%) | 13/98 (13.3%) | 10/57 (17.5%) | 5/42 (11.9%) | |
| Groin | 40/195 (20.5%) | 22/98 (22.4%) | 13/56 (23.2%) | 5/41 (12.2%) | |
| Any Patient Body site[a] | 51/197 (25.9%) | 27/98 (27.6%) | 16/57 (28.1%) | 8/42 (19.0%) | 0.52 |
| Bed control | 19/197 (9.6%) | 10/98 (10.2%) | 3/57 (5.3%) | 6/42 (14.3%) | |
| Call button | 15/197 (7.6%) | 9/98 (9.2%) | 6/57 (10.5%) | 0/42 (0) | |
| Table top | 15/197 (7.6%) | 7/98 (7.1%) | 8/57 (14.0%) | 0/42 (0) | |
| TV remote/buttons | 13/197 (6.6%) | 7/98 (7.1%) | 6/57 (10.5%) | 0/42 (0) | |
| Privacy curtain | 19/190 (10.0%) | 7/95 (7.4%) | 11/57 (19.3%) | 1/38 (2.6%) | |
| Bedrail | 20/197 (10.2%) | 12/98 (12.2%) | 6/57 (10.5%) | 2/42 (4.8%) | |
| Toilet seat | 19/197 (9.6%) | 10/98 (10.2%) | 8/57 (14.0%) | 1/42 (2.4%) | |
| Any Environmental site[b] | 57/197 (28.9%) | 31/98 (31.6%) | 19/57 (33.3%) | 7/42 (16.7%) | 0.14 |
| **RGNB** | | | | | |
| Nares | 4/197 (2.0%) | 1/98 (1.0%) | 3/57 (5.3%) | 0/42 (0) | |
| Hand | 2/197 (1.0%) | 0/98 (0) | 1/57 (1.8%) | 1/42 (2.4%) | |
| Groin | 24/195 (12.3%) | 9/98 (9.2%) | 10/56 (17.9%) | 5/41 (12.2%) | |
| Any Patient Body site[a] | 27/197 (13.7%) | 10/98 (10.2%) | 12/57 (21.1%) | 5/42 (11.9%) | 0.15 |
| Bed control | 4/197 (2.0%) | 1/98 (1.0%) | 3/57 (5.3%) | 0/42 (0) | |
| Call button | 1/197 (0.5%) | 0/98 (0) | 0/57 (0) | 1/42 (2.4%) | |
| Table top | 2/197 (1.0%) | 1/98 (1.0%) | 0/57 (0) | 1/42 (2.4%) | |
| TV remote/buttons | 1/197 (0.5%) | 0/98 (0) | 0/57 (0) | 1/42 (2.4%) | |
| Privacy curtain | 2/190 (1.1%) | 1/95 (1.1%) | 1/57 (1.8%) | 0/38 (0) | |
| Bedrail | 3/197 (1.5%) | 0/98 (0) | 2/57 (3.5%) | 1/42 (2.4%) | |
| Toilet seat | 4/197 (2.0%) | 1/98 (1.0%) | 3/57 (5.3%) | 0/42 (0) | |
| Any Environmental site[b] | 12/197 (6.1%) | 4/98 (4.1%) | 7/57 (12.3%) | 1/42 (2.4%) | 0.063 |

Some denominators are less than expected due to missing baseline samples, e.g., two participants were missing a groin sample at baseline. P-values (two-sided without adjustment for multiple comparisons) are based on Pearson's chi-square test.

MDRO multidrug-resistant organism, MRSA methicillin-resistant *Staphylococcus aureus*, RGNB resistant gram-negative bacteria, VRE vancomycin-resistant enterococci.

[a]Number of participants colonized at any of the three body sites sampled (nares, hand, or groin).

[b]Number of participants with environmental contamination at any of the seven environmental sites sampled (bed control, call button, table top, TV remote, privacy curtain, bedrail, or toilet seat).

**Table 3 | Association between participant characteristics and baseline MDRO colonization or new MDRO acquisitions**

| Characteristic | Unadjusted OR (95% CI) | P-value | Adjusted OR (95% CI) | P-value |
|---|---|---|---|---|
| **MDRO Colonization at Baseline (N = 188)[a]** | | | | |
| Recent Antibiotic Use | | | | |
| No (59) | 1 [Reference] | | 1 [Reference] | |
| Yes (129) | 3.45 (1.65-7.25) | <0.001 | 3.45 (1.53-7.81) | 0.003 |
| Wounds at Baseline | | | | |
| No (124) | 1 [Reference] | | 1 [Reference] | |
| Yes (64) | 2.00 (1.07-3.72) | 0.029 | 1.51 (0.75-3.04) | 0.243 |
| Katz (ADL) Score (188) | 1.13 (0.98-1.31) | 0.098 | 1.18 (0.99-1.39) | 0.051 |
| Preadmission Hospitalization Stay | | | | |
| ≤14 Days (142) | 1 [Reference] | | 1 [Reference] | |
| >14 Days (46) | 3.09 (1.56-6.14) | 0.001 | 2.94 (1.42-6.07) | 0.004 |
| NH Facility | | | | |
| Facility A (94) | 1 [Reference] | | 1 [Reference] | |
| Facility B (55) | 1.43 (0.72-2.83) | 0.302 | 2.09 (0.94-4.61) | 0.069 |
| Facility C (39) | 0.73 (0.32-1.64) | 0.442 | 0.82 (0.33-2.00) | 0.662 |
| **Any New MDRO Acquisition (N = 173)[b]** | | | | |
| Length of Stay in Study (days) (173) | 1.02 (1.01-1.03) | 0.006 | 1.02 (1.01-1.03) | 0.006 |
| NH Facility | | | | |
| Facility A (88) | 1 [Reference] | | 1 [Reference] | |
| Facility B (51) | 1.37 (0.68-2.76) | 0.379 | 1.18 (0.57-2.43) | 0.661 |
| Facility C (34) | 1.32 (0.59-2.34) | 0.503 | 1.47 (0.64-3.38) | 0.359 |

P-values (two-sided without adjustment for multiple comparisons) are based on logistic regression.
ADL activities of daily living, CI confidence interval, MDRO multidrug-resistant organism, NH nursing home, OR Odds ratio.
[a]Excludes 9 participants for which some data was missing: race/ethnicity (n = 7 missing) and hospital length of stay (n = 2 missing).
[b]Excludes 8 participants (race/ethnicity (n = 7 missing) and hospital length of stay (n = 1 missing)) and 16 participants for whom new acquisition could not be assessed (12 discharged after baseline visit and 4 colonized with all MDROs at baseline).

MDRO (40.9% of 181 at-risk), with an average time to new acquisition being 14.7 days (range, 1-65 days) (Table 4). The average time to new acquisition across all three facilities for MRSA, VRE, and RGNB, at any participant body site, was: 16.9 days (range, 5-42 days); 16.2 days (range, 1-89 days); and 19.4 days (range, 5-65 days), respectively. Participants colonized with an MDRO at a particular site at baseline were not considered at-risk of new acquisition for that MDRO at that site, which is why denominators are different at each site in Table 4. For the "any participant body site" category, participants who were colonized with all three MDROs at baseline were not considered at-risk of acquiring a new MDRO−this was the case for four participants. Rates of new acquisition at Facility A were: MRSA (8.1% of 86 at-risk participants), VRE (32.4% of 68 at-risk), RGNB (16.9% of 83 at-risk), and any MDRO (39.1% of 92 at-risk). Rates of new acquisition at Facility B were: MRSA (14.6% of 48 at-risk participants), VRE (30.0% of 40 at-risk), RGNB (16.3% of 43 at-risk), and any MDRO (44.2% of 52 at-risk). Rates of new acquisition at Facility C were: MRSA (11.8% of 34 at-risk participants), VRE (23.3% of 30 at-risk), RGNB (25.0% of 32 at-risk), and any MDRO (40.5% of 37 at-risk). New MDRO acquisitions during the NH stay were higher than baseline colonization across all body sites and environmental surfaces (Tables 2 and 4). Increased length of stay in the study was a risk factor for new MDRO acquisitions, suggesting ongoing transmission and more opportunities to acquire a new MDRO (Table 3).

### Transmissions during interactive visits

Study staff attended 345 total interactive visits with 135 study participants, comprised of: 209 (60.6%) physical (PT) and/or occupational therapy (OT) visits, 43 (12.5%) dining room visits (for recreation and/or dining), 35 (10.1%) radiation/oncology, 16 (4.6%) dialysis, 10 (2.9%) ophthalmology, and 8 (2.3%) radiology visits. Each visit was approximately 32 minutes (SD = 23.0) on average. A total of 58 transmissions (of either MRSA, VRE, or RGNB) occurred during 55 interactive visits

from 41 participants−19 transmissions of MRSA, 37 transmission of VRE, and 2 transmissions of RGNB.

The 55 interactive visits with at least one transmission event included 35 of 209 (16.7%) PT and/or OT visits, eight of 43 (18.6%) dining room visits, four of 16 (25.0%) dialysis sessions, and three of eight (37.5%) radiology visits. At Facility A, 38/231 (16.4%) interactive visits involved transmission of at least one MDRO, most commonly at radiology (3/8, 37.5%), dialysis (4/12, 33.3%), and PT/OT combined sessions (2/5, 40.0%). At Facility B, 14/59 (23.7%) interactive visits involved transmission of at least one MDRO, most commonly at OT (1/3, 33.3%) or other appointments such as speech pathology or resident care in their room (4/11, 36.3%). At Facility C, 3/55 (5.4%) interactive visits involved a transmission of at least one MDRO, most commonly at OT (2/14, 14.2%) or PT (1/17, 5.8%) (Table 5). Ten percent (35/345) of all interactive visits involved transmission of an MDRO to a surface; 6.6% (23/345) of interactive visits involved transmission of an MDRO to a participant hand.

Because RGNB transmissions were rare, we focus next on MRSA and/or VRE transmissions. Fifty-six transmissions of MRSA and/or VRE occurred during the 53 interactive visits from 39 participants (16 visits involved MRSA transmission only; 34 visits involved VRE transmission only; and 3 visits involved transmission of both VRE and MRSA). Figure 3 describes interactive visits in which MRSA and/or VRE is transmitted either to or from patient hands or surfaces, relative to participant hand colonization or surface contamination preceding the transmission. Ten interactive visits involved a patient hand becoming MRSA-positive, while 11 interactive visits involved a surface becoming MRSA-positive. Thirteen interactive visits involved a patient hand becoming VRE-positive, while 25 interactive visits involved a surface becoming VRE-positive. One visit with a MRSA transmission to a surface had no participant hand swabs collected and is not included in part a of Fig. 3. Transmission to equipment surfaces was more likely when participant hands were colonized with

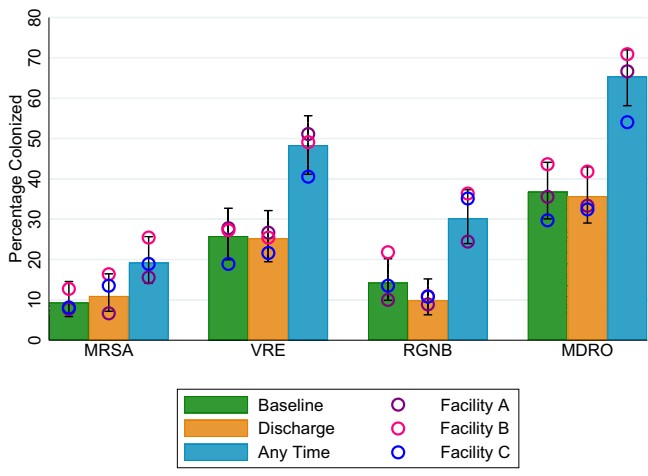

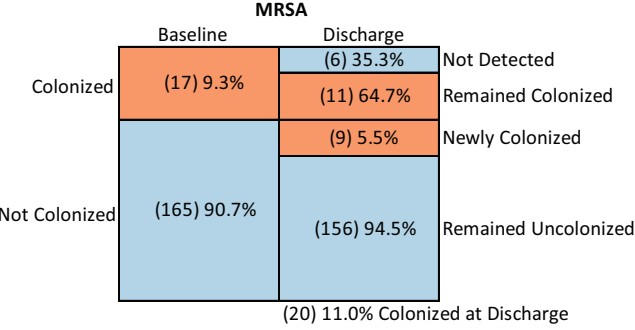

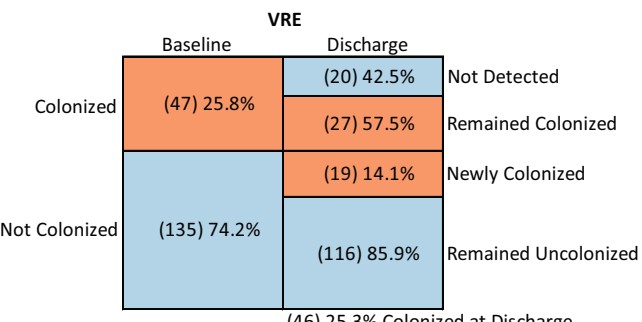

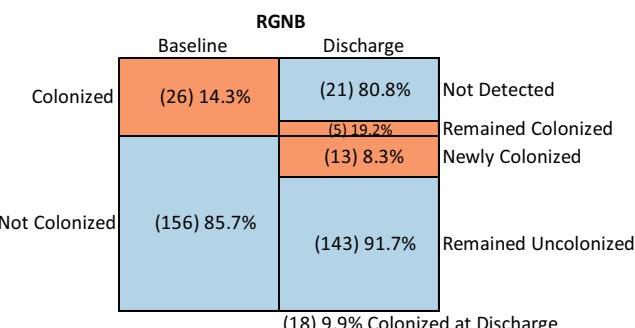

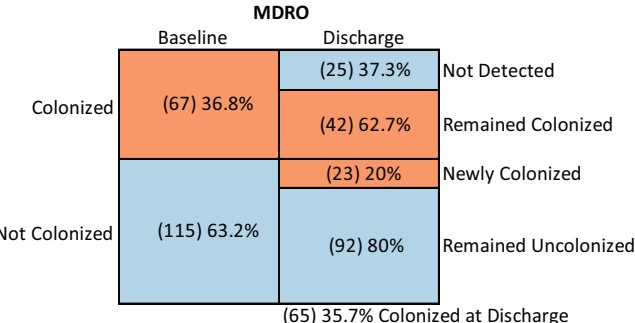

**Fig. 1 | Colonization with multidrug-resistant organisms at baseline, at discharge, and at any time during the study.** Bars indicate proportion of total participants colonized at baseline (green), discharge (orange), and any time (blue), among participants with >1 in-room visit (*n* = 182 participants), allowing distinct admission and discharge samples. Dots indicate proportion of participants at Facility A (purple, *n* = 90 participants), Facility B (pink, *n* = 55 participants), and Facility C (blue, *n* = 37 participants) colonized at baseline (green), discharge (orange), and any time (blue), among participants with >1 in-room visit (*n* = 182 participants). Error bars represent the 95% confidence interval for overall colonization (%), which is the average colonization rate among all three facilities. Based on Pearson's chi-square tests, no significant differences across the facilities were found. Source data are provided for this figure. Abbreviations: MDRO, multidrug-resistant organism; MRSA, methicillin-resistant *Staphylococcus aureus*; RGNB, resistant gram-negative bacilli; VRE, vancomycin-resistant enterococci.

MRSA or VRE at the visit start than when they were not. For example, of the 36 interactive visits where the participant hand was contaminated with VRE before use (Fig. 3c), transmission of VRE to a surface occurred at 16 visits, for a transmission rate of 44.4%; whereas, of the 35 interactive visits where equipment or surface was contaminated with VRE before use (Fig. 3d), transmission of VRE to the participant's hand occurred at 2 visits, for a transmission rate of 5.7%. Equipment contamination with MRSA or VRE was not associated with transmission to participant hands (Fig. 3).

### Integration of Surveillance and Sequencing Data

To confirm sources of transmission events during interactive visits, we focused our genomic analysis on a subset of participants for whom sequencing was completed and whom had a transmission occur during an interactive visit (i.e., participants in categories 4b and 4c of Supplementary Table 3). This analysis includes 338 isolates (125 MRSA and 213 VRE) from 14 participants with at least one transmission of VRE or MRSA identified during one or more interactive visits. Among these 14 participants, 23 transmission events were identified over 19 visits. Forty-eight percent of these transmission events had a likely source of transmission based on microbiology tests (11 transmissions during nine visits from seven participants) while 52% had an unknown source of transmission (12 transmissions during 10 visits from nine participants). Two participants had instances of both unknown and known sources (See Supplementary Fig. 5 for further details on these two participants).

Among the 11 transmissions with a known or suspected source identified by microbiology results, WGS confirmed identical strains between transmission source and destination surface 100% (11/11) of the time. These 11 transmissions with a known or suspected source involved the following organisms and directions: VRE was transmitted

**Fig. 2 | Changes in MDRO colonization status from baseline visit to discharge (last study visit).** Orange boxes indicate participants colonized with an MDRO, blue boxes indicate participants not colonized with an MDRO. Numbers in each box indicate the number of participants with a given colonization status at study baseline or discharge. For example, 47 participants who had multiple in-room visits were colonized with VRE at baseline. Of those, 27 (57.5%) remained colonized at discharge. The remaining 20 (42.5%) did not have detectable colonization at discharge. Alternatively, 135 participants who had multiple in-room visits were not colonized with VRE at baseline. Of those, 19 (14.1%) acquired VRE during their NH stay, while 116 (85.9%) remained not colonized with VRE. Source data are provided for this figure. Abbreviations: MDRO, multidrug-resistant organism; MRSA, methicillin-resistant *Staphylococcus aureus*; NH, nursing home; RGNB, resistant gram-negative bacilli; VRE, vancomycin-resistant enterococci.

**Table 4 | New acquisition of MDROs on participant and in participant rooms during the study period**

| Any MDRO | No. Participants with new colonization or contamination / total participants at-risk (%) | | | | |
|---|---|---|---|---|---|
| | All facilities | Facility A | Facility B | Facility C | P-value |
| Nares | 35/185 (18.9%) | 11/93 (11.8%) | 17/55 (30.9%) | 7/37 (18.9%) | |
| Hand | 55/185 (29.7%) | 28/93 (30.1%) | 19/55 (34.5%) | 8/37 (21.6%) | |
| Groin | 61/185 (33.0%) | 32/93 (34.4%) | 16/55 (29.1%) | 13/37 (35.1%) | |
| Any Patient Body site[a] | 74/181 (40.9%) | 36/92 (39.1%) | 23/52 (44.2%) | 15/37 (40.5%) | 0.84 |
| Bed control | 34/185 (18.4%) | 14/93 (15.1%) | 15/55 (27.3%) | 5/37 (13.5%) | |
| Call button | 32/185 (17.3%) | 16/93 (17.2%) | 11/55 (20.0%) | 5/37 (13.5%) | |
| Table top | 39/185 (21.1%) | 18/93 (19.4%) | 14/55 (25.5%) | 7/37 (18.9%) | |
| TV remote/buttons | 49/185 (26.5%) | 25/93 (26.9%) | 17/55 (30.9%) | 7/37 (18.9%) | |
| Privacy curtain | 53/185 (28.7%) | 21/93 (22.6%) | 25/55 (45.5%) | 7/37 (18.9%) | |
| Bedrail | 46/185 (24.9%) | 18/93 (19.4%) | 23/55 (41.8%) | 5/37 (13.5%) | |
| Toilet seat | 76/185 (41.1%) | 36/93 (38.7%) | 28/55 (50.9%) | 12/37 (32.4%) | |
| Any Environmental site[b] | 88/181 (48.6%) | 44/89 (49.4%) | 33/55 (60.0%) | 11/37 (29.7%) | 0.017 |
| **MRSA** | | | | | |
| Nares | 14/174 (8.1%) | 4/87 (5.0%) | 9/53 (17.0%) | 1/34 (2.9%) | |
| Hand | 19/174 (10.9%) | 6/88 (6.8%) | 8/51 (15.7%) | 5/35 (14.3%) | |
| Groin | 9/176 (5.1%) | 5/91 (5.5%) | 3/51 (5.9%) | 1/34 (2.9%) | |
| Any Patient Body site[a] | 18/168 (10.7%) | 7/86 (8.1%) | 7/48 (14.6%) | 4/34 (11.8%) | 0.50 |
| Bed control | 10/172 (5.8%) | 3/87 (3.4%) | 4/50 (8.0%) | 3/35 (8.6%) | |
| Call button | 8/175 (4.6%) | 3/89 (3.4%) | 3/51 (5.9%) | 2/35 (5.7%) | |
| Table top | 12/175 (6.9%) | 5/89 (5.6%) | 4/51 (7.8%) | 3/35 (8.6%) | |
| TV remote/buttons | 14/176 (8.0%) | 7/89 (7.9%) | 6/53 (11.3%) | 1/34 (2.9%) | |
| Privacy curtain | 14/171 (8.2%) | 5/87 (5.7%) | 6/51 (11.8%) | 3/33 (9.1%) | |
| Bedrail | 13/174 (7.5%) | 4/88 (4.5%) | 7/52 (13.5%) | 2/34 (5.9%) | |
| Toilet seat | 20/182 (11.0%) | 6/91 (6.6%) | 11/55 (20.0%) | 3/36 (8.3%) | |
| Any Environmental site[b] | 28/162 (17.3%) | 12/83 (14.5%) | 13/47 (27.7%) | 3/32 (9.4%) | 0.067 |
| **VRE** | | | | | |
| Nares | 13/176 (7.4%) | 6/89 (6.7%) | 5/50 (10.0%) | 2/37 (5.4%) | |
| Hand | 35/159 (22.0%) | 20/81 (24.7%) | 12/46 (26.1%) | 3/32 (9.4%) | |
| Groin | 37/145 (25.5%) | 20/71 (28.2%) | 9/42 (21.4%) | 8/32 (25.0%) | |
| Any Patient Body site[a] | 41/138 (29.7%) | 22/68 (32.3%) | 12/40 (30.0%) | 7/30 (23.3%) | 0.67 |
| Bed control | 26/165 (15.8%) | 11/82 (13.4%) | 14/52 (26.9%) | 1/31 (3.2%) | |
| Call button | 24/170 (14.1%) | 12/84 (14.3%) | 9/49 (18.4%) | 3/37 (8.1%) | |
| Table top | 29/168 (17.3%) | 12/85 (14.1%) | 12/46 (26.1%) | 5/37 (13.5%) | |
| TV remote/buttons | 35/171 (20.5%) | 18/85 (21.2%) | 11/49 (22.5%) | 6/37 (16.2%) | |
| Privacy curtain | 42/161 (26.1%) | 18/83 (21.7%) | 19/45 (42.2%) | 5/33 (15.2%) | |
| Bedrail | 33/165 (20.0%) | 13/81 (16.0%) | 17/49 (34.7%) | 3/35 (8.6%) | |
| Toilet seat | 53/166 (31.9%) | 31/81 (38.3%) | 15/48 (31.3%) | 7/37 (18.9%) | |
| Any Environmental site[b] | 61/132 (46.2%) | 32/63 (50.8%) | 22/38 (57.9%) | 7/31 (22.6%) | 0.008 |
| **RGNB** | | | | | |
| Nares | 11/179 (6.2%) | 2/90 (2.2%) | 5/52 (9.6%) | 4/37 (10.8%) | |
| Hand | 9/183 (4.9%) | 5/93 (5.4%) | 1/54 (1.9%) | 3/36 (8.3%) | |
| Groin | 23/158 (14.6%) | 13/83 (15.7%) | 5/44 (11.4%) | 5/31 (16.1%) | |
| Any Patient Body site[a] | 29/158 (18.4%) | 14/83 (16.9%) | 7/43 (16.3%) | 8/32 (25.0%) | 0.55 |
| Bed control | 2/179 (1.1%) | 0/90 (0) | 1/52 (1.9%) | 1/37 (2.7%) | |
| Call button | 2/182 (1.1%) | 2/91 (2.2%) | 0/55 (0) | 0/36 (0) | |
| Table top | 9/180 (5.0%) | 4/90 (4.4%) | 5/54 (9.3%) | 0/36 (0) | |
| TV remote/buttons | 2/182 (1.1%) | 1/91 (1.1%) | 1/55 (1.8%) | 0/36 (0) | |
| Privacy curtain | 6/176 (3.4%) | 1/88 (1.1%) | 5/54 (9.3%) | 0/34 (0) | |
| Bedrail | 7/181 (3.9%) | 1/91 (1.1%) | 5/54 (9.3%) | 1/36 (2.8%) | |
| Toilet seat | 12/179 (6.7%) | 4/90 (4.4%) | 6/52 (11.5%) | 2/37 (5.4%) | |
| Any Environmental site[b] | 26/172 (15.1%) | 9/87 (10.3%) | 13/49 (26.5%) | 4/36 (11.1%) | 0.031 |

Some denominators are less than expected due to missing baseline samples, e.g., two participants were missing a groin sample at baseline. P-values (two-sided without adjustment for multiple comparisons) are based on Pearson's chi-square test.

MDRO multidrug-resistant organism, MRSA methicillin-resistant *Staphylococcus aureus*, RGNB resistant gram-negative bacteria, VRE vancomycin-resistant enterococci.

[a]Number of participants colonized at any of the three body sites sampled (nares, hand, or groin).

[b]Number of participants with environmental contamination at any of the seven environmental sites sampled (bed control, call button, table top, TV remote, privacy curtain, bedrail, or toilet seat).

**Table 5 | Transmission prevalence among various interactive visits attended**

| Interactive Visit Type | No. visits with a transmission event / no. visits attended (%) | | | | | | | | | | | |
| | All Facilities | | | Facility A | | | Facility B | | | Facility C | | |
| | All transmissions | Transmission to surface | Transmission to participant hand | All transmissions | Transmission to surface | Transmission to participant hand | All transmissions | Transmission to surface | Transmission to participant hand | All transmissions | Transmission to surface | Transmission to participant hand |
|---|---|---|---|---|---|---|---|---|---|---|---|---|
| All therapy visits | 35/209 (16.7%) | 22/209 (10.5%) | 16/209 (7.6%) | 26/140 (18.5%) | 17/140 (12.1%) | 12/140 (8.5%) | 6/23 (26.0%) | 5/23 (21.7%) | 1/23 (4.3%) | 3/46 (6.5%) | 0/46 (0) | 3/46 (6.5%) |
| Physical Therapy | 23/138 (16.6%) | 14/138 (10.1%) | 11/138 (7.9%) | 17/101 (16.8%) | 10/101 (9.9%) | 9/101 (8.9%) | 5/20 (25.0%) | 4/20 (20.0%) | 1/20 (5.0%) | 1/17 (5.8%) | 0/17 (0) | 1/17 (5.8%) |
| Occupational Therapy | 10/51 (19.6%) | 6/51 (11.8%) | 5/51 (9.8%) | 7/34 (20.5%) | 5/34 (14.7%) | 3/34 (8.8%) | 1/3 (33.3%) | 1/3 (33.3%) | 0/3 (0) | 2/14 (14.2%) | 0/14 (0) | 2/14 (14.2%) |
| PT and OT combined | 2/20 (10.0%) | 2/20 (10.0%) | 0/20 (0) | 2/5 (40%) | 2/5 (40.0%) | 0/5 (0) | - | - | - | 0/15 (0) | 0/15 (0) | 0/15 (0) |
| NH Dining Room | 8/43 (18.6%) | 4/43 (9.3%) | 4/43 (9.3%) | 4/21 (19.0%) | 2/21 (9.5%) | 2/21 (9.5%) | 4/22 (18.1%) | 2/22 (9.0%) | 2/22 (9.0%) | - | - | - |
| Radiation/Oncology | 0/35 (0) | 0/35 (0) | 0/35 (0) | 0/34 (0) | 0/34 (0) | 0/34 (0) | - | - | - | 0/1 (0) | 0/1 (0) | 0/1 (0) |
| Dialysis | 4/16 (25.0%) | 2/16 (12.5%) | 2/16 (12.5%) | 4/12 (33.3%) | 2/12 (16.6%) | 2/12 (16.6%) | 0/1 (0) | 0/1 (0) | 0/1 (0) | 0/3 (0) | 0/3 (0) | 0/3 (0) |
| Eye clinic/Ophthalmology | 1/10 (10.0%) | 1/10 (10.0%) | 0/10 (0) | 1/7 (14.2%) | 1/7 (14.2%) | 0/7 (0) | 0/2 (0) | 0/2 (0) | 0/2 (0) | 0/1 (0) | 0/1 (0) | 0/1 (0) |
| Radiology (e.g., x-ray, CT scan) | 3/8 (37.5%) | 2/8 (25.0%) | 1/8 (12.5%) | 3/8 (37.5%) | 2/8 (25.0%) | 1/8 (12.5%) | - | - | - | - | - | - |
| Prosthetics/Orthotics | 0/6 (0) | 0/6 (0) | 0/6 (0) | 0/6 (0) | 0/6 (0) | 0/6 (0) | - | - | - | - | - | - |
| Other[a] | 4/18 (22.2%) | 4/18 (22.2%) | 0/18 (0) | 0/3 (0) | 0/3 (0) | 0/3 (0) | 4/11 (36.3%) | 4/11 (36.3%) | 0/11 (0) | 0/4 (0) | 0/4 (0) | 0/4 (0) |
| Total[b] | 55/345 (15.9%) | 35/345 (10.1%) | 23/345 (6.6%) | 38/231 (16.4%) | 24/231 (10.3%) | 17/231 (7.3%) | 14/59 (23.7%) | 11/59 (18.6%) | 3/59 (5.0%) | 3/55 (5.4%) | 0/55 (0) | 3/55 (5.4%) |

NH nursing home, OT occupational therapy, PT physical therapy.

[a]Includes infrequent visits to podiatry, orthopedics, speech pathology, urology, vascular, cardiology, and miscellaneous care in the participant's room.

[b]Multiple transmissions are possible within a single visit.

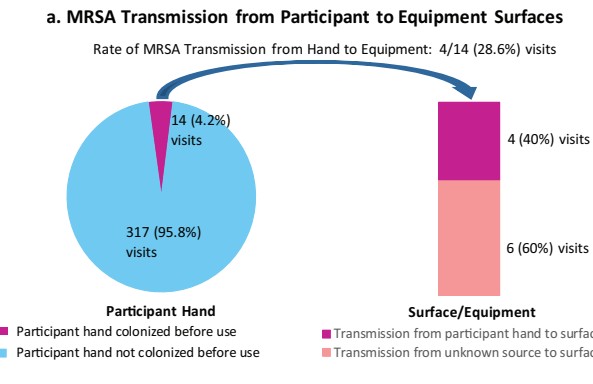

### a. MRSA Transmission from Participant to Equipment Surfaces

Rate of MRSA Transmission from Hand to Equipment: 4/14 (28.6%) visits

14 (4.2%) visits
317 (95.8%) visits
**Participant Hand**
- Participant hand colonized before use
- Participant hand not colonized before use

4 (40%) visits
6 (60%) visits
**Surface/Equipment**
- Transmission from participant hand to surface
- Transmission from unknown source to surface

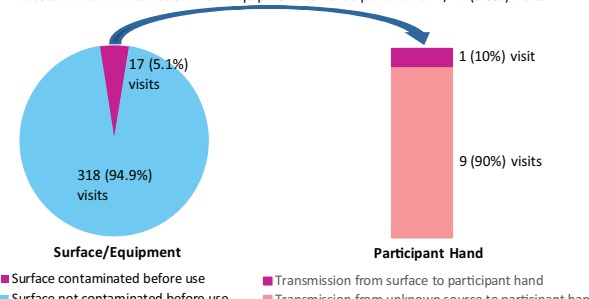

### b. MRSA Transmission from Equipment Surfaces to Participant

Rate of MRSA Transmission from Equipment to Participant Hand: 1/17 (5.9%) visits

17 (5.1%) visits
318 (94.9%) visits
**Surface/Equipment**
- Surface contaminated before use
- Surface not contaminated before use

1 (10%) visit
9 (90%) visits
**Participant Hand**
- Transmission from surface to participant hand
- Transmission from unknown source to participant hand

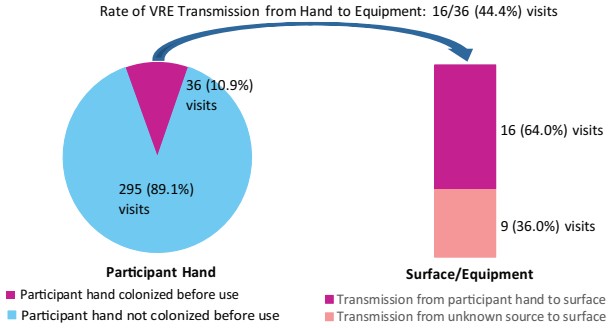

### c. VRE Transmission from Participant to Equipment Surfaces

Rate of VRE Transmission from Hand to Equipment: 16/36 (44.4%) visits

36 (10.9%) visits
295 (89.1%) visits
**Participant Hand**
- Participant hand colonized before use
- Participant hand not colonized before use

16 (64.0%) visits
9 (36.0%) visits
**Surface/Equipment**
- Transmission from participant hand to surface
- Transmission from unknown source to surface

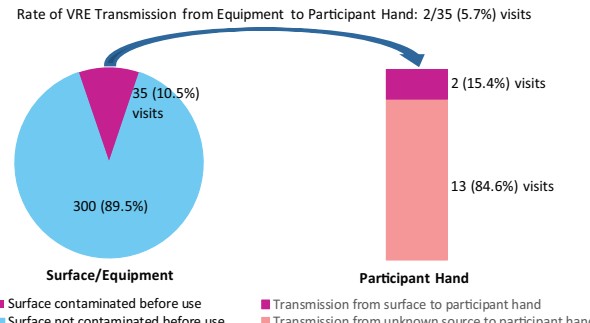

### d. VRE Transmission from Equipment Surfaces to Participant

Rate of VRE Transmission from Equipment to Participant Hand: 2/35 (5.7%) visits

35 (10.5%) visits
300 (89.5%)
**Surface/Equipment**
- Surface contaminated before use
- Surface not contaminated before use

2 (15.4%) visits
13 (84.6%) visits
**Participant Hand**
- Transmission from surface to participant hand
- Transmission from unknown source to participant hand

### e. Unadjusted Odds Ratios

| MRSA | OR (95% CI) | P value |
|---|---|---|
| **Transmission to Equipment** | | |
| Participant hand not colonized first | 1 [Reference] | |
| Participant hand colonized first | 20.7 (5.1-84.7) | 2.5e-5 |
| **Transmission to Participant** | | |
| Equipment not contaminated first | 1 [Reference] | |
| Equipment contaminated first | 2.2 (0.3-18.3) | 0.451 |
| **VRE** | **OR (95% CI)** | **P value** |
| **Transmission to Equipment** | | |
| Participant hand not colonized first | 1 [Reference] | |
| Participant hand colonized first | 21.3 (8.2-55.6) | 4.1e-10 |
| **Transmission to Participant** | | |
| Equipment not contaminated first | 1 [Reference] | |
| Equipment contaminated first | 1.7 (0.4-7.7) | 0.505 |

**Fig. 3 | Transmission of DNA-confirmed MRSA or VRE to participant hands or equipment. a** MRSA transmission to equipment. Of the 14 interactive visits where the participant hand was contaminated with MRSA before use, transmission of MRSA to a surface occurred at 4 (28.6%) visits. Ten MRSA transmissions to a surface occurred, of which 4 (40%) can be attributed to participant hand contamination. **b** MRSA transmission to participant hand. Of the 17 interactive visits where equipment or surface was contaminated with MRSA before use, transmission of MRSA to the participant's hand occurred at 1 (5.9%) visit. Ten MRSA transmissions to a participant hand occurred, of which 1 (10%) can be attributed to surface contamination. **c** VRE transmission to equipment. Of the 36 interactive visits where the participant hand was contaminated with VRE before use, transmission of VRE to a surface occurred at 16 (44.4%) visits. Twenty-five VRE transmissions to a surface occurred, of which 16 (64.0%) can be attributed to participant hand contamination. **d** VRE transmission to participant hand. Of the 35 interactive visits where equipment or surface was contaminated with VRE before use, transmission of VRE to the participant's hand occurred at 2 (5.7%) visits. Thirteen VRE transmissions to a

participant hand occurred, of which 2 (15.4%) can be attributed to surface contamination. **e** Unadjusted odds ratios and confidence intervals for the association between surface contamination and participant hand colonization and transmission of MRSA and VRE. Reported *p*-values (two-sided) are based on generalized linear models specifying an exchangeable, within-participant correlation structure for the panels. Transmission of MRSA to a surface/equipment was 20.7 times more likely when the participant hand was colonized first, compared to when the participant hand is not colonized first ($p = 2.5e-5$), while transmission of MRSA to a participant hand is 2.2 times more likely when the surface/equipment is contaminated first ($p = 0.451$). Transmission of VRE to a surface/equipment was 21.3 times more likely when the participant hand was colonized first, compared to when the participant hand is not colonized first ($p = 4.1e-10$), while transmission of VRE to a participant hand is 1.7 times more likely when the surface/equipment is contaminated first ($p = 0.505$). Source data are provided for this figure. Abbreviations: MRSA, methicillin-resistant *Staphylococcus aureus*; VRE, vancomycin-resistant enterococci; OR, odds ratio.

from the patient's hand to a surface 9 times; and MRSA was transmitted from the patient's hand to a surface 2 times. Fig. 4a shows one specific example of a VRE transmission event to a surface (circled in red) with a known or suspected source (i.e., the participant's hand), confirmed via sequencing results.

Among the 12 transmissions with no microbiologically detected source during the interactive visit, WGS confirmed the presence of an identical strain found on the participant or in their room at an earlier visit 66.7% (8/12) of the time. These 12 transmissions with an unknown source involved the following organisms and directions: VRE was

a.

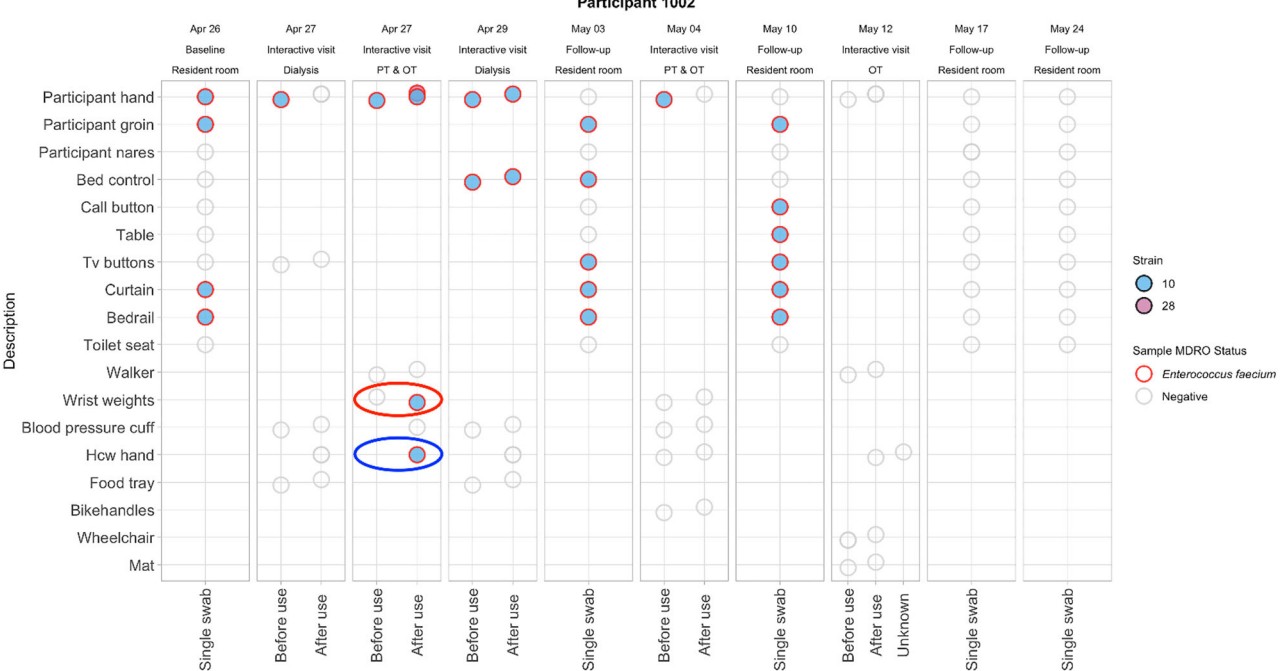

b.

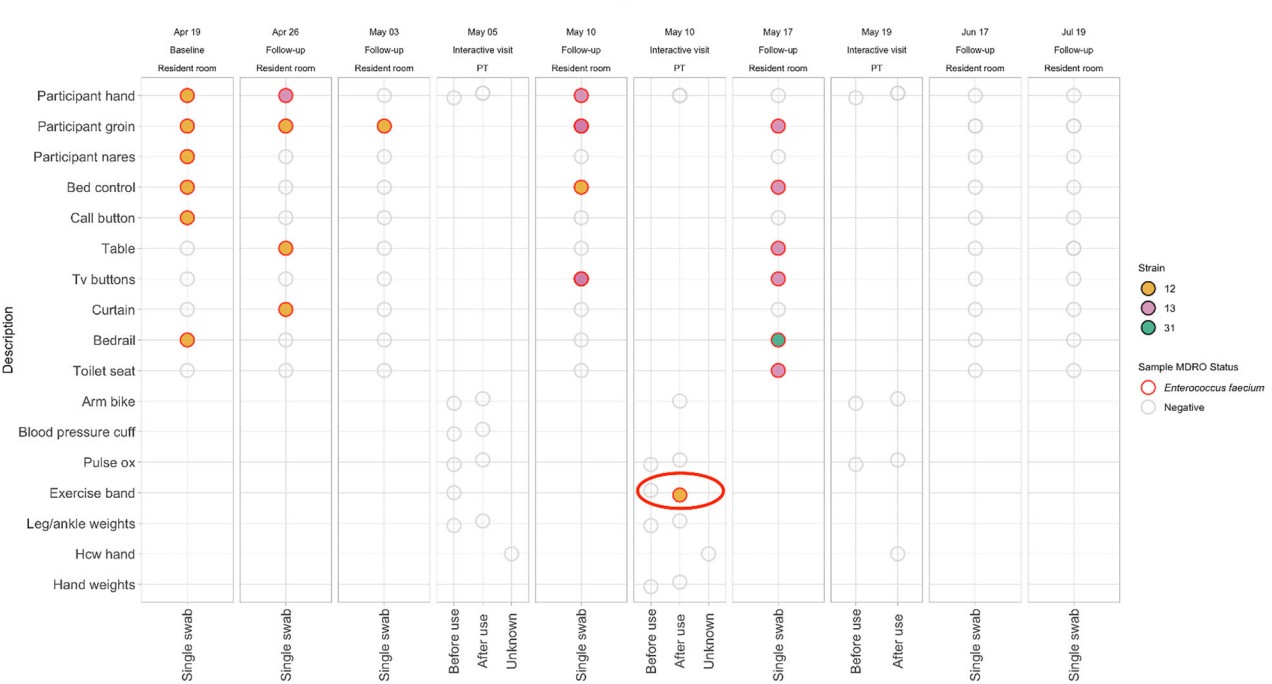

transmitted to patient's hand two times; VRE was transmitted to a surface two times; MRSA was transmitted to a patient's hand four times; and MRSA was transmitted to a surface four times. Figure 4b shows one specific example of a VRE transmission event to a surface (circled in red) from an unknown source. Sequencing results enable the linkage of this transmission to the participant's hand, colonized with the same strain at an earlier, in-room visit.

Additionally, there were seven instances among these 14 participants where transmission could not be assessed because either a "before" or an "after" swab were not collected (e.g., a HCP hand was swabbed only one time, identified as the blue circle in Fig. 4a). For these seven instances (six participants during six visits), WGS confirmed an identical strain of either VRE or MRSA on the participant or in their room at an earlier visit 85.7% (6/7) of the time, suggesting these were not new acquisitions.

**Fig. 4 | Genomics support of microbiological linkages among two participants.** Column headings indicate date, location, and type (baseline or follow-up = in-room; or interactive) of each study visit. Transmission events are identified with a red circle; single positive swabs not able to be assessed for transmission are identified with a blue circle. White circles outlined in gray are samples collected and negative for any MDRO; colored circles outlined in red are samples collected and positive for a particular strain (listed in each legend). **a** For participant 1002, VRE strain 10 was detected at multiple body sites (groin and hand) and environmental surfaces (bedrail and privacy curtain) at study baseline (Apr 26). Study staff attended a dialysis session with this participant on Apr 27, during which time the participant's hand was colonized with VRE strain 10 at the start of the session, but no transmission of VRE to any surfaces was detected during that interactive visit. Study staff also attended a PT & OT combined session on that same day (Apr 27); during this interactive visit, a VRE strain 10 transmission (circled in red) was detected at the wrist weights, since the weight went from VRE-negative to VRE-positive following participant use. Based on our microbiology results, the participant's hand is the likely source of this transmission, since the participant's had was VRE-positive at the beginning of the interactive visit. Furthermore, sequencing confirms these VRE strains are identical. The healthcare provider hand at the end of the session was also VRE-positive; however, this is not counted as a transmission because we do not have a "before" swab on the healthcare provider hand to prove that it changed status from negative to positive. Sequencing results confirm this strain is identical to that carried on the participant hand, highly suggestive of transmission. This participant continued to be colonized, and their in-room surface contaminated, at two subsequent in-room visits. This participant's hand was also colonized during two subsequent interactive visits, but no other transmissions were detected. **b** For participant 1001, VRE strain 12 was detected at multiple body sites (nares, groin, and hand) and environmental surfaces (bed control, bedrail, call button) at study baseline (on Apr 19). During the first in-room follow-up visit (Apr 26), VRE strain 12 was detected at the participant groin, table top, and privacy curtain, while VRE strain 13 was detected on the participant's hand. During the next in-room follow-up visit (May 3), VRE strain 12 was detected only at the participant groin. Study staff next attended a PT session with this participant (May 5), during which time no MDROs were found on the participant's hand, on surfaces, nor on the healthcare provider hand. The next visit with this participant was in his or her room (May 10), where VRE strain 13 was detected at the participant's groin, hand, and TV remote, while VRE strain 12 was detected at the bed control. Study staff attend a PT session with this participant next (on May 10), during which time a VRE strain 12 transmission (circled in red) was detected at the exercise band, since the exercise band changed status from VRE-negative to VRE-positive following participant use. The participant's hand was not positive for VRE at the start of this session, so we cannot assume the source of this transmission was the participant's hand. However, sequencing allows us to speculate that the participant's hand is the likely source, since their hand was colonized with VRE strain 12 at an earlier, in-room visit (on Apr 19). This participant continued to be colonized (VRE strain 13), and their in-room surface contaminated (VRE strains 13 and 31), at one subsequent in-room visit. This participant's hand was not colonized at any subsequent interactive visits. Abbreviations: MDRO, multidrug-resistant organism; OT, occupational therapy; PT, physical therapy; VRE, vancomycin-resistant enterococci.

## Discussion

In this prospective, longitudinal cohort study of nearly 200 residents newly admitted to three VA NHs in the Midwestern United States, over one-third of participants were colonized with at least one MDRO upon admission and over 40% acquired a new MDRO during their stay. Further, over one-third of those colonized at baseline were no longer colonized at discharge. In this study, researchers followed NH residents during interactive visits to the therapy gym, dialysis, and other hospital venues to understand spread of MDROs. Our study reveals that transmission of MDROs is common, occurring during one in six (16.5%) such visits.

The multivariate logistic analyses conducted for this study confirm previously reported risk factors associated with increased risk of MDRO colonization for community NH residents, including prolonged hospitalization (>14 days), need for assistance with ADLs, and recent antibiotic exposure (within the previous 30 days)[13–17]. However, the need for assistance with ADLs was not statistically significant in our risk factor analysis. We found that longer stays within the VA NH also correlate with increased risk of new MDRO acquisition, with VRE the most frequently acquired organism. Similar studies in VA NHs have focused largely on MRSA. Stone et al.[18] noted that 22% of newly admitted residents were colonized with MRSA. For the additional 10% of VA NH residents who acquired MRSA colonization over a six-month observation period, antibiotic use was the only identified risk factor. In our study, while the proportion of MRSA colonization among newly admitted VA NH residents was lower (8.6%), we found a similar proportion of residents with newly acquired MRSA (9.2%). The higher proportion of VRE and RGNB colonization at baseline and acquisition while in the VA NH may be due to differences in the inherent transmissibility of these organisms or may reflect the long-term effects of the enterprise-wide MRSA surveillance policy implemented by the VA[19,20]. Spontaneous loss of MDRO colonization among our cohort confirms the dynamic nature of bacterial colonization[13,21,22].

We found that participant hands frequently transmitted MRSA or VRE, and less frequently RGNB, to environmental surfaces outside of their rooms. Several reasons could explain this observation. First, participant hands were less likely to harbor RGNB[13,23]. Second, RGNB are more transient and less subject to shedding[24,25]. Third, it is possible that new acquisition of RGNB may not be from environmental surfaces but from other mechanisms, including the hands of healthcare workers or conversion of a sensitive GNB to a resistant GNB after exposure to antibiotics[26].

Study staff spent over 10,000 minutes attending interactive visits, swabbing participant hands, HCP hands, and surfaces while simultaneously and meticulously recording details of each swab and interaction. Transmission events occurred most during therapy visits− 35 transmissions occurred during 209 therapy sessions attended (17%, for physical or occupational therapy). Gontjes et al.[27] attended ten rehabilitation sessions at two community NHs. Defining transmission the same as in the present study, they found in 35 opportunities for transmission, microorganism transmission occurred 17.1% of the time. Gontjes et al. also found the most contaminated equipment to be the arm bike handle (17.0%), pulley (11.8%), stairs (10.2%), and mat (8.5%). Therapy gyms are dynamic spaces often shared between various HCPs as well as inpatient and outpatient populations. These and other commonly shared spaces, such as dialysis units and dining rooms, are critical areas to focus on improved resident hand hygiene and consistent cleaning practices. Li et al.[28] examined MDRO transmission from ten residents in a single VA NH, recovering MRSA from 41% (26/41) of surfaces contacted by NH residents known to be colonized with MRSA. In our study, among participants colonized with MRSA at baseline, 94% (16/17) had their immediate environment contaminated with MRSA and 29% (5/17) transmitted MRSA to a surface during an interactive appointment.

Our findings should be interpreted in the context of the following important limitations. First, it is not possible to exclude the presence of false negatives. However, our culture-based methods involving consistent sampling of a relatively large surface area with flocked swabs, as well as enrichment, provide for a high sensitivity for all target organisms, while maintaining a specificity for live, transmission-capable organisms, unlike non-culture-based techniques. Second, while our analysis considered MRSA and VRE as distinct organisms each with a single resistance phenotype, we did not conduct in depth analysis on individual RGNB species and their transmission. Our study was designed to assess transmission of bacteria responsible for healthcare-acquired infections to and from VA NH residents, their environment, and HCPs. Third, while some participants seemingly lost colonization with the MDROs assessed, both VRE and many species of RGNB may remain in the gastrointestinal tract for extended periods, with antibiotic exposures leading to increased numbers of these

organisms. When MDROs are no longer detected via bacterial culture, the risk of transmission is greatly decreased; however, subsequent new self-colonization from MDROs originating in the gastrointestinal tract is always a possibility. Fourth, while we used a protocol ensuring high sensitivity to detect each target species in each sample, it is possible that within a single sample, multiple strains of the same species could be present at the same time, some of which could be missed because of phenotypically indistinguishable colonies. Fifth, since this study focused on antimicrobial resistant organisms (MRSA and VRE), transmission of methicillin-susceptible *S. aureus* (MSSA) and vancomycin-susceptible *Enterococcus* (VSE) has not been investigated. Our focus on antibiotic-resistant pathogens may underestimate transmission of various other pathogens occurring during interactive visits. Last, recruitment and visit frequencies across the three facilities was unequal, a consequence of COVID-19 restrictions and policy variations across the sites. Likewise, only three NHs were involved out of the 132 VA NHs across the nation; thus, the results may not be generalizable to other VA NHs or to non-VA NHs. However, the fact that we were able to implement a complex project at more than one facility simultaneously is a strength, as most literature in this area describes a single facility.

This study also has several strengths. First, this study is sizable in its implementation—no other study in the VA NH setting examines MDRO transmission at visits outside the participant's room. Second, we made multiple visits to participants inside and outside their rooms, which helps elucidate the trajectories and colonization dynamics at critical points of the participants' stay. Finally, we swabbed multiple anatomic sites which, coupled with extensive clinical metadata collected by trained research staff, provides a unique comprehensive picture of participant colonization with MDROs. Our study is a critical step towards developing interventions that target and reduce transmission occurring during out-of-room interactive visits.

In summary, MDRO prevalence on admission to NHs is high, new acquisition is common, and residents are frequently discharged to the community with MDROs. Transmission of MDROs during interactive visits is common, especially among residents with hand colonization. Residents with long prior hospitalizations and other risk factors are more likely to be colonized with MDROs, suggesting that a subpopulation of residents should be prioritized for MDRO preventive efforts that engage residents, prescribers, and frontline personnel, paying particular attention to transmissions occurring outside of resident rooms.

## Methods
### Study population and design
The Movement And Transmission of Resistant and InfXous organisms (MATRIX) study was a multisite, prospective cohort study conducted at three VA NHs (Ann Arbor, MI, Detroit, MI, and Cleveland, OH) from April 2021 to September 2023. The three NHs varied in size, with the number of skilled nursing beds for A, B, and C being 46, 174, and 115 beds, respectively. See Supplementary Table 4 for detailed facility characteristics and infection control policies collected during interviews with infection control leadership.

Eligibility criteria for study participation included new admission to the NH and enrollment within three days, with written informed consent and Health Insurance Privacy and Portability Act (HIPAA) authorization obtained from the resident and/or their legally authorized representative. Exclusion criteria were non–English speaking and receipt of hospice care. We followed enrolled participants for up to three months or until discharge, whichever came first. Two different types of visits were completed with each participant: 1) regularly-scheduled in-room visits, which were conducted on the day of study enrollment, weekly for one month, and monthly thereafter for a maximum of three months (i.e., days 0 (baseline), 7, 14, 21, 30, 60, 90); and 2) up to five interactive visits, including but not limited to therapy

gym visits for physical or occupational therapy, dialysis unit visits, radiology visits for x-rays, ophthalmology visits for routine eye appointments, radiation/oncology visits, and dining room visits for meals or recreation/activities. The timing of interactive visits was dependent on the participant's scheduled appointments with these services. The types of swabs collected during in-room visits and interactive visits are described in the Microbiologic Methods section below and in Supplementary Tables 5 and 6.

This study complies with all relevant ethical regulations, including the collection of written informed consent from all participants. The study was reviewed and approved by the VA Central Institutional Review Board (cIRB) and all three participating local site research oversight boards. Sequence data that support the findings of this study have been deposited in the National Center for Biotechnology Information (NCBI) BioSample database, with Bioproject number PRJNA1204323. The source data underlying Figures and Supplementary Figs. are provided as Source Data File. Additional details on datasets (such as aggregated data) and protocols that support findings of this study will be made available by the corresponding author upon reasonable written request, in accordance with our VA-funded data management and access plan and with appropriate permissions from the VA Central Review Board. The authors will give feedback within 30 days.

### Clinical data collection
Demographic data (including age, sex, race/ethnicity) and risk factors for MDRO colonization (including functional disability, use of indwelling devices, presence of comorbidities, open wounds, prior antibiotic use, history of infection(s), and length of hospitalization) were collected by trained research staff who reviewed data and clinical notes stored in the electronic health record for each enrolled participant at study baseline and discharge. We recorded functional disability per occupational and physical therapy assessments using the Katz scale, which scores dependence as either zero (dependent) or one (independent) in six categories of self-maintenance (bathing, dressing, toileting, transferring, continence, and feeding), with the total score being the sum of six possible dependencies[29,30]. Device use was defined as having a feeding tube, indwelling urinary catheter, or peripherally inserted central catheter (PICC) in place at the time of study enrollment. Wounds were defined as open if they required regular dressing changes or a wound vacuum; this included purulent wounds, vascular ulcers, diabetes-related ulcers, open surgical wounds, or pressure ulcers greater than stage one. We used the Charlson Comorbidity Index as a summary comorbidity measure[31]. Prior antibiotic exposure was defined as documented receipt of an antibiotic within 30 days of study enrollment[13]. We defined clinical infections as the presence of both a clinical note in the participant's medical record and the prescription of a systemic antibiotic in the past 30 days[14] and prolonged hospital stay prior to VA NH admission as >14 days.

### Microbiologic methods
No disinfection policies nor hand hygiene access were modified because of participation in this study. During in-room baseline and follow-up visits, staff collected microbiological samples from the participant's hands (the entire palm of both hands and fingertips), nares, and groin, and from seven environmental surfaces in the participant's room (swabbing an area of approximately 5x20cm, when available): bed controls, bed rail, call button, television (TV) remote, bedside tabletop, privacy curtain, and toilet seat. During unscheduled interactive visits, staff collected microbiological samples from participant hand(s) before and after surface or equipment contact, from surfaces or equipment before and after participant contact, and from the hand(s) of HCPs interacting directly with the participant at least once. All participant hand, surface, and HCP hand interactions were recorded, including the timing and order in which they occurred. Specimens

were collected using sterile flocked swabs in transport media (Eswabs, BD, Cat. N. 220245, Franklin Lakes, NJ) and assessed for MDROs including methicillin-resistant *Staphylococcus aureus* (MRSA), vancomycin-resistant enterococci (VRE), and resistant gram-negative bacilli (RGNB) in our research laboratory using previously described, standard microbiologic methods and appropriate quality controls[13,14]. Specifically, 0.2 mL of the participant hand and environmental surface swab transport medium were enriched in Brain Heart Infusion Broth (BD DIFCO, Cat. N. 244100 Franklin lakes, NJ) overnight at 36 °C, and the enriched culture subsequently streaked onto Mannitol Salt Agar (MSA, Remel, Cat. N. R01580, Lenexa, KS), Bile Esculin Agar (BD, Cat. N. 212205, Franklin Lakes, NJ) with vancomycin (Chem-Impex Cat. N. 00315, Wood Dale, IL) 6 mg/L (BEV6) agar, and MacConkey agar (Remel, Cat. N. R01552, Lenexa, KS) for isolation of MRSA, VRE, and gram-negatives. The swab transport medium from patient nares and groin samples was streaked directly to MSA, BEV6 and MacConkey plates. Colonies potentially suggestive of *S. aureus* were identified using catalase and coagulase test (Staphaurex, Cat. N. R30859901ZL30, Remel, Lenexa, KS). *S. aureus* isolates were then tested for resistance (MRSA) using disc-diffusion cefoxitin (Thermo Fisher, Cat. N. CT0119B, Waltham, MA) resistance in Mueller-Hinton Agar (BD, Cat. N. 225250, Franklin Lakes, NJ). Colonies suggestive of VRE on BEV6 were confirmed with pyrrolidonyl arylamidase testing (DrySlide, BD, Cat. N. 23174760, Franklin Lakes, NJ). All gram-negatives were identified using API 20E strips (Biomerieux, Cat. N. 20160, France), and tested for resistance to ceftazidime, ceftazidime/clavulanic acid, fluoroquinolones and carbapenems using disc-diffusion (BD BBL, Cat. Ns 231754, 231753, 231704, 231658, Franklin Lakes, NJ) in Mueller-Hinton Agar. All MDROs were frozen in BHI/25% glycerol for subsequent whole genome sequencing of strains involved in potential transmission events. We classified gram-negative bacilli as RGNB if they were resistant to any one of the following antibiotics: ceftazidime (30 µg), ceftazidime and clavulanic acid (30/10 µg), ciprofloxacin (5 µg), or meropenem (10 µg).

## Whole genome sequencing

To identify isolates of interest for whole genome sequencing (WGS), we prioritized participants into distinct categories (see Supplementary Table 3) indicating colonization during in-room visits and transmission during interactive visits. In total, we processed 438 microbiologically defined VRE and MRSA isolates (238 VRE, 200 MRSA) from 27 unique participants for sequencing, of which 409 isolates from 26 patients passed QC and were confirmed as vancomycin-resistant *Enterococcus faecium* (166 isolates), *Enterococcus faecalis* (46 isolates), *Enterococcus gallinarium* (11 isolates) or MRSA (186 isolates). Criteria for passing QC included: i) depth of sequencing > 40X, ii) a genome assembly of <500 contigs and iii) genome size between 2.5 Mb and 3.5 Mb.

We sent frozen bacterial isolates of interest to the University of Michigan Microbiome Core, where DNA extractions (DNA preparation and quantification) took place on Epimotion liquid handling robots using the QIAGEN MagAttract microbial DNA kit. Genomic libraries were prepared with the QIASeq FX DNA library prep kit and sequenced at the University of Michigan Advanced Genomics Core on an Illumina NovaSeq 6000, with 150-bp paired-end reads. Raw sequencing reads were trimmed using Trimmomatic[32] v.0.39 to remove adapters and low-quality bases. Trimmed high-quality reads were assembled using Spades[33] v.3.15.3, annotated using Prokka[34] v.1.14.5, and had MLST assigned using the MLST tool[35]. Trimmed sequencing reads were then mapped to species-specific reference genomes (*E. faecium* -Aus0004/ GenBank CP003351.1, *E. faecalis* - V583/GenBank NC_004668.1, MRSA - USA300_TCH1516/ NC_010079.1) using the Burrows–Wheeler Aligner-MEM[36] v.0.7.17 and variants were called and filtered using Samtools v.1.11[37]. Variants were filtered from raw results using GATK's VariantFiltration (QUAL, >100; MQ, >50; >=10 reads supporting variant;

and FQ, <0.025). In addition, a custom python script was used to filter out single- nucleotide variants that were: (i) <5 base pairs (bp) in proximity to indels that were identified by GATK HaplotypeCaller, (ii) in a phage region identified by Phaster[38] or (iii) they resided in tandem repeats of length greater than 20 bp as determined using the exact-tandem program in MUMmer[39]. Only variants located at nucleotide positions present in all isolates (i.e., core positions) and which did not have a variant filtered in any isolate were considered for the transmission analyses. The total portion of the reference genomes considered for transmission analysis were 77% (2,199,367 / 2,872,915) for *E. faecium*, 83% (2,677,413 / 3,218,031) for *E. faecalis* and 87% (2,547,304 / 2,872,915) for *S. aureus*. SNV thresholds to determine putative cross-transmission were determined based on analysis of within-resident genetic diversity (see Supplementary Fig. 6), based on the premise that the amount of diversity within a patient sets the upper bound on the SNV distance expected between direct transmission pairs[40]. To this end, histograms of SNV distances for intra- and inter-resident pairs were examined (see Supplementary Fig. 6), with thresholds for VRE (10 SNVs) and MRSA (20 SNVs), below which 98% and 95% are intra-resident pairs, and being consistent with previous thresholds reported using epidemiologic data[41,42]. These thresholds were used to group isolates into strain groups, based on the criteria that all members of the group be within the species-specific threshold to each other. Whole-genome phylogenies overlaid with participant ID and strain groups show that while strain groups are typically unique to a participant's isolates, there are instances of likely transmission where another participant isolate is embedded within a participant's isolates (Supplementary Fig. 7).

## Main outcomes and definitions

The primary study outcomes were: 1) prevalence of longitudinal changes in MDRO colonization, including new acquisitions and loss of MDROs; 2) MDRO transmission events during various interactive visits, including identification of interaction types associated with higher transmission rates; and 3) utilization of WGS to confirm sources of MDRO transmission during interactive visits.

For outcome one, a participant was considered colonized at baseline if an MDRO (MRSA, VRE or RGNB) was detected at any one of the three participant body sites sampled (i.e., nares, groin, or hand). Similarly, if an MDRO (MRSA, VRE or RGNB) was detected at any one of the seven environmental surfaces sampled (i.e., bed controls, bed rail, call button, television (TV) remote, bedside tabletop, privacy curtain, or toilet seat), the participant's environment was considered contaminated with that MDRO. We use "colonization" to refer to the participant, and "contamination" to refer to the environment. A participant was considered colonized at discharge if an MDRO (MRSA, VRE or RGNB) was detected at any one of the three participant body sites sampled (i.e., nares, groin, or hand) on the day of the last in-room study visit. We used the entire sample of study participants (*n* = 197, Tables 1 & 2) as well as a subset of participants with >1 in-room visit completed (*n* = 182, Figs. 1 & 2) to analyze MDROs present on baseline and at discharge (i.e., the last in-room study visit). New acquisition was defined as the patient colonization or environmental contamination of an MDRO (MRSA, VRE, or RGNB) at any follow-up visit (in-room or interactive visits), which was not detected at the baseline study visit (day 0). The acquisition rate was defined as new acquisition events per 1000 resident-days. New acquisition was assessed at the participant level (e.g., a participant culture negative for MRSA at baseline with a subsequent follow-up or interactive visit culture positive for MRSA at any anatomic site). To estimate new MDRO acquisition during the NH stay, we limited the analysis to participants with at least one follow-up in-room or interactive visit after baseline (*n* = 185, Table 4). For the acquisition analysis by specific MDRO (MRSA, VRE, or RGNB), we further excluded participants colonized at baseline with that MDRO at each site sampled. For the acquisition analysis of all MDROs, we

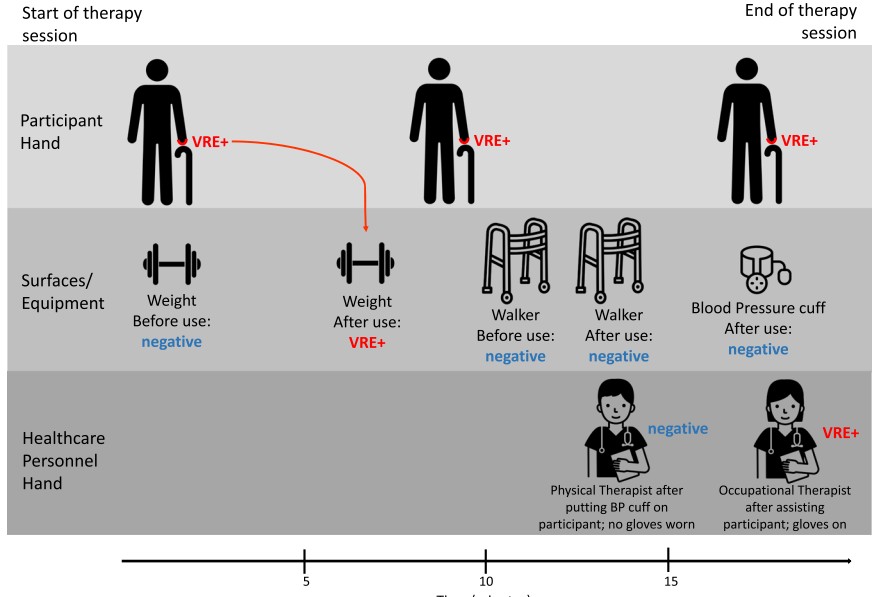

**Fig. 5 | A representative interactive visit that resulted in a transmission event.** This participant, who was colonized with VRE at the hand, visited the rehabilitation gym for physical and occupational therapy. The first piece of equipment used by the participant was a hand weight; the hand weight was negative for any MDRO prior to use and was contaminated with VRE after approximately 6 minutes of participant contact. This change in surface contamination is an example of a transmission event, with the source being the participant's hand and the destination surface being the weight. The participant's hand remained colonized with VRE during the session. The walker was used next by the participant -- it was negative for any MDRO prior to use and after two minutes of use, no transmission occurred. The last item of contact, the blood pressure cuff, was not swabbed before use, but was negative for any MDRO after being on the participant's arm for five minutes. The bare hand of the physical therapist was negative for any MDRO after taking the participant's blood pressure. The occupational therapist's gloved hand was positive for VRE at the end of the session, following several touches to the participant and walker. Because we only swabbed the occupational therapist's hand one time, we cannot conclusively say a transmission occurred (as we do not know if their hand was negative at the session start). The participant's hand remained colonized with VRE at the end of the session. "Nurse" icon by Llisole from https://thenounproject.com/browse/icons/term/nurse/ CC BY 3.0. All other icons downloaded from the-nounproject.com via a paid subscription (no attribution required).

excluded those colonized at baseline with all three MDROs (excluded 4 participants colonized with all three MDROs at baseline).

For outcome two, we assessed transmission events during observed interactive visits only (up to five per participant) and used $n = 135$ participants who had at least one interactive visit completed (Table 5 & Fig. 3). Transmission was defined as a change in microbial colonization or contamination from negative to positive following use or interaction, when swabs were collected *both* before and after the interaction, within a single interactive visit. Transmission events were only assessed during interactive visits (not in-room visits), as swabs could feasibly be collected *both* before use/interaction and after use/interaction. Figure 5 provides an example of an interactive visit that revealed transmission of VRE from the participant hand to a weight used during a physical therapy session. In further describing transmission events, we use the term "source" to describe the surface or hand that is contaminated first, and "destination surface" to describe the surface or hand which becomes contaminated (i.e., changes from negative to positive) following interaction with the source.

For outcome three, we used WGS to establish whether isolates involved in transmission events shared a clonal origin. We used $n = 14$ participants who had at least one transmission event occur during an interactive visit and had sequencing completed (Fig. 4). We considered strains to be linked by a putative transmission event if they were within 10 single nucleotide variants (SNVs) for VRE and within 20 SNVs for MRSA, based on distributions of pairwise distances within and between participants. We report frequencies for the following three scenarios: (1) identical strains found at the source and destination surface; (2) when the source is unknown, identical strains found at the destination surface and on the participant or in their room at an earlier visit; and (3) when a "before" or "after" sample is missing and

transmission is not able to be assessed, identical strains found at a contaminated site and on the participant or in their room at an earlier visit.

## Statistical analysis

Participant characteristics were described both for the overall cohort and stratified by facility (Table 1). We examined risk factors for baseline MDRO colonization and subsequent acquisition of new MDROs using logistic regression models. Covariates for the model were selected based on a univariate screening process (Supplementary Tables 1 and 2), with a more liberal selection threshold of $p = 0.1$. This threshold is more inclusive than the $p = 0.05$ commonly used for hypothesis testing, allowing for the identification of potentially important covariates that might otherwise be overlooked. Both univariate (Supplementary Tables 1 and 2) and multivariable (Table 3) analyses were conducted using participants with complete data for all risk factors.

Of the 197 residents initially included in the study, data on baseline MDRO colonization were available for all, while data on race/ethnicity and hospital length of stay (LOS) were missing for 7 (3.6%) and 2 (1.0%) residents, respectively. Complete data were available for 188 (95.4%) residents (Supplementary Table 1). For the analysis of new MDRO acquisition, 181 residents who were not colonized with all three MDROs upon enrollment were included in the analysis. Among these, data on race/ethnicity were missing for 7 (3.9%) and on hospital LOS for one resident, yielding complete data for 173 (95.6%) residents (Supplementary Table 2).

To address missing data, we implemented the multivariable logistic regression model using the initial study samples and applied full information maximum likelihood (FIML) estimation. FIML avoids listwise deletion and uses all available data points to complete the analysis[43]. The

results from FIML estimation were consistent with those from complete case analysis; therefore, we report estimates based on the latter.

For visit-level data obtained from interactive visits, we applied generalized estimating equation (GEE) models with an exchangeable correlation structure to account for the potential correlations among repeated measurements[44]. This approach allowed us to explore two main relationships: 1. the association between contamination of participant hands and subsequent transfer of contaminants to equipment surfaces, and 2. the association between equipment contamination and the transfer of contaminants back to participant hands.

The statistical significance was determined at 2-sided $P < .05$, and given the exploratory nature of our study, $p$-values were reported without adjustment for multiple hypotheses tested. All statistical analyses were performed using Stata 17 software (StataCorp. 2021. Stata Statistical Software: Release 17. College Station, TX: StataCorp LLC) and Mplus software version 8.8 (Muthén & Muthén).

### Reporting summary
Further information on research design is available in the Nature Portfolio Reporting Summary linked to this article.

### Data availability
Sequence data that support the findings of this study (corresponding to Fig. 4 and Supplementary Figs. 5–7) have been deposited in the National Center for Biotechnology Information (NCBI) BioSample database, with Bioproject number PRJNA1204323. The source data underlying Figs. 1, 2, & 3 and Supplementary Figs. 2–4 are provided as Source Data File. Additional details on datasets (such as aggregated data) and protocols that support findings of this study will be made available by the corresponding author (Lona Mody, lonamody@med.umich.edu) upon written request, in accordance with our VA-funded data management and access plan and with appropriate permissions from the VA Central Review Board. The authors will give feedback within 30 days. This publication made use of the PubMLST website (https://pubmlst.org/) developed by Keith Jolley *(Jolley & Maiden 2010, BMC Bioinformatics, 11:595)* and sited at the University of Oxford. The development of that website was funded by the Wellcome Trust. Source data are provided with this paper.

### Code availability
No custom code was developed in this study.

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

## Acknowledgements

We extend our appreciation to leadership, staff, and residents at all participating VA sites. This work was supported by VA CSRD Merit Review Grant (5I01CX001691) and the Veterans Affairs (VA) Ann Arbor Healthcare System, John D. Dingell VA Medical Center, Northeast Ohio Healthcare System, and the Pittsburgh Healthcare System. Dr. Mody is supported by the National Institute on Aging Mid-Career Mentorship Grant (K24 AG050685); Michigan Institute for Clinical and Health Research (UL1TR002240); the Claude D. Pepper Older Americans Independence Center (P30 AG024824); and the Agency for Healthcare Research & Quality (Grant RO1HS25451). Funding from co-authors includes: NIH T32GM141746 and a University of Michigan Rackham Merit Fellowship (Clement); National Institutes of Allergy and Infectious Diseases (R01AI175227, Snitkin); and VA HSR Research Career Scientist award (RCS 11-222, Krein).

## Author contributions

Conceptualization: L.M. (Mody), S.S., S.L.K., M.R.J., L.M. (Min), T.G., M.R.; Data curation: K.G., M.C., G.V., T.C., J.T., J.R., A.N., O.H., T.B.; Formal analysis: G.V., A.G., T.G.; Funding acquisition: L.M. (Mody), S.S., S.L.K., M.R.J., L.M. (Min), T.G., M.R.; Investigation: K.G., M.C., G.V., T.C., J.T., J.R., A.N., O.H., T.B.; Methodology: K.G., M.C., G.V., T.C., S.S., S.L.K., M.R.J., J.T., J.R., A.N., O.H., T.B.; Resources: K.G., M.C., G.V., T.C., J.T., J.R., A.N., O.H., T.B.; Software: G.V., A.G.; Supervision: L.M. (Mody), K.G., L.C., F.P., R.L.P.J.; Visualization: L.M. (Mody), K.G., M.C., G.V., T.C., E.S., A.G.; Writing – original draft: L.M. (Mody), K.G., M.C., G.V.; Project administration, Validation, and Writing – review & editing: L.M. (Mody), K.G., M.C., G.V., T.C., E.S., S.S., S.L.K., M.R.J., J.T., J.R., A.N., O.H., T.B., N.G.E.C., L.M. (Min), A.G., T.G., M.R., L.C., F.P., R.L.P.J.

## Competing interests

The authors declare no competing interests.
