## [Transparent Peer Review file · Nature Communications]

Epidemiology and transmission dynamics of multidrug-resistant organisms in nursing homes within the United States

Corresponding Author: Dr Lona Mody

Version 0:

Reviewer comments:

Reviewer #1

(Remarks to the Author)

General Comments:

This prospective cohort evaluation of MDRO transmission at 3 nursing homes is a unique addition to the literature as the investigators are able to describe and measure the transmission dynamics within the nursing home setting between patient/surfaces/staff. The whole genome sequencing analysis adds another fascinating layer of complexity to the review of transmission patterns showing how multiple strains can be involved in acquisition. The authors have presented the rationale and methods in a very clear manner. The overall results are clearly stated, however, some of the specifics within the tables/figures could be even more impactful with some clarification and even by condensing the number of tables/figures (see below for specific suggestions).

Specific Comments:

Abstract

1. Lines 49-50, we would recommend adding details to include environmental sampling of surfaces and hands as these represent a great deal of work and the key findings of the paper.

Methods

2. Lines 109... , please include a statement about whether disinfection protocols or access to hand hygiene were modified as a result of participation and knowledge about this study.

3. Lines 117-118, please include details about what surface area of the hands were swabbed.

4. Line 120, please comment on any known information about the sensitivity of the environmental and hand sampling. This information is needed for better interpretation of the acquisition/loss findings and should be included in a short statement in the discussion.

Results

5. Please reconcile the number of participants included between Supplemental Tables 2-3 and Table 1 and with the enrollment numbers in Supplemental Figure 2. Simple footnotes about these differences at the bottom of Supplemental Tables 2-3 would be helpful.

6. We find Figure 4 to be more informative and a better representation of the main findings of the study compared to Figure 2. You may consider replacing Figure 2 with Figure 4.

7. Line 224-225, the higher ADL scores are not overwhelming based on the magnitude of the OR and the 95% CI overlapping the null value. You may consider not highlighting that prominently in the text or providing more rationale for highlighting this finding. This point is also referred to in line 281.

8. Figure 5, even with careful reading of the legend and axes, we think this figure is difficult to interpret as currently presented. The readers will be quick to think these % represent how often equipment was not contaminated vs.

contaminated rather than % of transmission. This figure is so central to the main unique findings of this paper it may be worth exploring if there is a clearer way to present it.

9. Line 256, please specify which categories from Supplement Table 1 were used to select the 14 participants for sequencing.

10. If space for figures is limited, Figures 3 and supplemental Figures 3 may not be needed beyond including the main findings directly in the text.

Discussion

11. Lines 279-282 mentions risk factors that Table 2 did not show to be statistically significant findings. Please delineate between your study's findings and those that you are referencing in the literature.

12. Line 323, Please consider also mentioning whether you believe your findings to be generalizable to non-VA nursing home patients.

Reviewer #2

(Remarks to the Author)

This important work by Mody and colleagues is a welcome addition to our understanding of colonization, transmissions, and infections with drug-resistant bacterial pathogens in the NH setting. The focus on interactive visits paired with patient and room environment baseline of colonization is a noteworthy and underscores the potential of MDRO transmission during encounters outside the resident rooms. This was a multicenter cohort study of 3 VA NHs with nearly 200 participants included, providing ample data and analysis opportunities using complementary methods to address the study's primary objectives of quantifying MDRO prevalence, transmission events, and transmission dynamics. This is a novel study that represents a considerable effort to characterize carriage rates of important MDROs as well as their transmission within a complex and relatively understudied subpopulation of NH residents. I recommend this for publication following the clarification of these points:

The authors present detailed epidemiological data to make inferences about putative transmission events, however it is unclear if the WGS data was fully used to support the microbiologic tests representing likely transmission or those transmissions that did not have any detected sources. Considering the effort of sequencing 338 isolates, why are there no species-specific whole-genome phylogenies or contact network analyses presented? In Figure 6, five *E. faecium* strains are presented in the two examples of putative transmission, yet phylogenetic and/or network context is missing to explain strain identification from the much larger pool of isolates and transmission events that are not presented.

Minor comments:

It is unclear how the species-specific SNV thresholds were calculated and what percentage of within-patient diversity these would cover? The cited reference to support lines 145-147 refers to a study of *C. difficile* where a threshold of 2 SNVs was implemented to identify recent transmissions; however the implementation of 10 and 12 SNVs for VRE and MRSA, respectively (lines 171-173), are different and based on data presented in Supplemental Figure 1, although these values are not sufficiently explained.

Figure 4: Decolonization implies active measures of IPC to remove target organisms – perhaps revise to “spontaneous decolonization” or “not detected”?

Reviewer #3

(Remarks to the Author)

Mody et al. presented the manuscript, EPIDEMIOLOGY OF MULTIDRUG-RESISTANT ORGANISMS (MDROS) ACROSS THREE VA NURSING HOMES AND MDRO TRANSMISSION DURING INTERACTIVE VISITS”. The authors conducted a multisite, prospective cohort study at three VA nursing homes and used genomics to examine colonization and potential transmission of MRSA, VRE, and R-GNB during activities at the nursing homes in a subset of 27 individuals. A major strength of the manuscript is the study design which examined 3 facilities. My first major concern with this manuscript is that it is currently very difficult to read – 1) it is written as if it were a clinical trial, but according to the reporting summary it is not and 2) the current display items are of interest to a very narrow audience interested in infection prevention. The strengths of the data were not adequately leveraged to answer questions about the extent to which we see differences in the pattern of transmission of MRSA, VRE or other pathogens across the 3 facilities or the types of interactive visits. Since patients were treated as a single population across the 3 facilities, the analysis is not reflective of the multi-site cohort design. Technically, I have concerns about the way colonization and transmission are defined. Likewise, the reader is unable to discern whether “transmission” frequencies appear to be greater for one species relative to others or with respect to the time interval separating sample collection (1 day vs. 1 or 3 months). Moreover, it is unclear as to whether we see multiple body sites of a single individual colonized by one pathogen or how often we see multiple pathogens colonizing a single individual. It is the opinion of this reviewer that for this manuscript to be of broad interest to people outside of infection control a major reworking of the analysis and the main display figures is essential. Since this is not a specialized journal focused on infection prevention or clinical trials, many of the current display items would be better suited to the supplement so that main figures related to the primary objectives – and the microbiology/genomic epidemiology -- can be included. The current analysis limits my enthusiasm for the manuscript. Finally, the methods are not sufficiently written – I would not be able to reproduce the genomic findings, for example, even given the data in SRA, and the reporting summary should be more comprehensively completed.

Major points

1. “We conducted a multisite, prospective cohort study at three VA NHs (Ann Arbor, MI, Detroit, MI, and Cleveland, OH) from April 2021 to September 2023.” It's important to note that in the main text and in the reporting summary the 3 cohorts are treated for the most part as if they belong to a single population. If the investigators wish to position this paper as a multisite prospective cohort study, then it's important to assess the magnitude of differences in the outcome variables based on the

facility in which each cohort was sampled. For this work to be considered of general interest to readers outside of infection prevention community the authors may want to consider reworking the paper to explore the cohort angle a bit.

a. The reader has to look at supplementary figure 4 to see the distinct environmental sites that were sampled. Given that differences in infection prevention practices (not just policies) could lead to different transmission dynamics it is suggested that these data for participant room in sFig4 be presented on a per-facility basis

b. It seems to me that a main finding of interest is whether the number of transmission events varied based on the type of interactive event, the time interval separating interactive events, and the facility in which the participant was located. I don't understand why this isn't presented as a main figure. Is it that regardless of interactive visit the feature determining transmission was hand colonization? For example, of the 17 visits where there was equipment contamination with MRSA and transmission, were they all different kinds of visits? Do we see higher frequency within vs. out of room?

2. Since we are talking about transmission and de-colonization events, the summary statistics about sample numbers is an insufficient way to describe the study design. It's important for the reader to know whether we're looking at multiple colonized body sites of one person, whether multiple species were found on the same person, as well as the number of time points over which we observe an apparent "colonization" or transmission event per person. The authors are encouraged to present a figure that displays subject, body site, and time point with information about the type of interactive visit and to present this information for each facility. Since it appears figures were made in R, this could be accomplished with some clever use of the facet grid function and the shape/color function of `geom_point`. This reviewer believes this is essential in the reformulation of this manuscript and necessary in the supplement.

a. Presentation of cohort design. The subjects who meet certain criteria for which transmission/de-colonization will be studied should be presented as well as an overview of the types of visits (frequency of each kind) should be presented.

i. It is important for the reader to understand how many samples of which variety the 185 participants contributed over time – as well as the number of individuals in each facility. This is to help the reader understand whether a more targeted analysis of a subset of subjects would better serve the questions posed.

ii. Likewise, the reader needs to understand how many samples of which variety the 134 participants contributed during the interactive visits.

3. Furthermore, since samples were collected on a daily basis for a month and on a monthly basis thereafter it's important for the analysis to report whether we different dynamics based on the interval separating sample collection. Do we see putative transmission events more often when we look at samples separated in time by 1 day vs. 2 or 7 days, and are we calling events transmission events when they are separated by more than 1 month? If transmissions are linked by a month or more, then describing the event as a transmission seems like it could be a bit of a stretch.

4. Were any of these individuals roommates and how did sharing a room impact transmission frequencies

5. Prevalence of colonization

a. First, how is colonization defined, given that multiple body sites were sampled? Please specify on line 216.

b. Second, with respect to defining colonization, how do we distinguish between contamination of a surface with an MDRO and colonization, which implies stable growth? Do you have instances in which you observe a "transmission" and subsequent evidence of growth at that same site?

c. Third, how do these patterns differ across the nursing home settings? If the collection of samples across facilities is important, then the data should be analyzed by facility, rather than aggregated as if the samples were all drawn from a single facility.

d. Finally, what is the limit of detection for each of the species that you're surveying? If you were to sequence samples, would you identify *S. aureus* (and other species) reads and/or the genes conferring abx resistance? Given that we're trying to understand transmission or strain "displacement", it's very important for the reader to believe that the colonization negatives are in fact true negatives.

e. With respect to colonization positivity/negativity, the reader needs to have a feel for the numbers of colonies for each species that was plated and sampled and identified. Otherwise, it is hard to understand whether had you sampled more colonies you'd see colonization at different time points. The number of colonies sampled for each species should be presented. If you were to do a rarefaction curve, would it be saturated?

f. Finally, how would these patterns change were you to have sampled all the isolates of a given species, not just the ones that were antibiotic resistant? It could be that the selection method gives us an under-estimate of colonization positivity (e.g., someone is colonized with *S. aureus* which acquires abx resistance). For this reason, it's important for the authors to spell out the methods used for selection of these species, rather than referring the reader to an external set of sources (lines 122-124). Certainly, there should be a limitations section that acknowledges that the study design may have led to an incomplete assessment of the prevalence of colonization as well as the appearance of acquisition events.

g. the authors should consider presenting an analysis of the pan genome relative to the core genome, rather than relying simply on SNPs, particularly since some of these SNP distributions are extraordinarily small. 20 SNPs isn't much genomic variation. Similarly, while the authors mention sequence typing was performed, I don't see these data presented anywhere

6. "Of these, four participants were colonized with all three MDROs at enrollment and thus not considered at risk of acquiring any new MDRO, resulting in 181 participants with 725 in-room follow-up visits." This reviewer would appreciate an analysis of the clones at baseline and at follow-up for the individuals who were colonized at enrollment. Did we see the same strain at follow up or do you believe there was a transmission and displacement event at follow-up? Do we see episodes of mixed colonization where two strains are present? Importantly, the time scale of sample collection is also important....

7. Participants frequently acquired a new MDRO: MRSA (10.7% of 168 at-risk participants), VRE (29.7% of 138 at-risk), R-GNB (18.4% of 158 at-risk), and any MDRO (40.9% of 181 at-risk), with an average time to new acquisition being 14.7 days (range, 1-65 days). For the reader to understand the average time to new acquisition we need to have a better understanding of the number of days separating sample collection for each individual sampled. I'm also wondering whether the average time to acquisition differed by pathogen and/or the facility.

8. With respect to figure 4, does this show colonization of any body site or colonization of a single body site? Please specify. If acquisition or loss at individual body sites per person is not shown, it should be in order for the reader to be convinced of acquisition or loss, particularly in light of concerns regarding the limit of detection.

9. Again, with respect to supplementary table 5, it's important for the reader to understand the distribution of interactive visits across time for each subject and whether or not a transmission event occurred. A presentation of this type of raw data is necessary for the reader to interpret the summary statistics presented in the main manuscript.

10. "Equipment contamination with MRSA and VRE, however, was not associated with transmission to participant hands. How long after touching the surface was the hand sampled?" Was there an attempt to see if over time you saw a different outcome?

11. The text between 255-268 is hard to follow. This text is written in a way that makes it hard to get the point through the details.

a. For example, here you say "Among the 12 transmissions with no microbiologically detected source during the interactive visit, WGS confirmed the presence of an identical strain found on the participant or in their room at an earlier visit 66.7% (8/12) of the time." If the strain was present on that individual, how is this considered a transmission event? It's important to distinguish between translocation from one body site to another and transmission from one individual or environmental source to another.

b. Similarly, "WGS confirmed identical strains between transmission source and destination surface 100% (11/11) of the time." Would be better described as the number of instances of transmission from an environmental source to a specific body site vs. other patterns of transmission.

c. Likewise, I'm getting lost in the numbers. There were 14 individuals for whom a transmission event was identified. 7 had a confirmed source of transmission. 9 had an unknown source of transmission. This is 16 individuals. This implies that 2 individuals had both a known and an unknown source of transmission. That's an important detail that gets lost in the current representation of the data. The authors are encouraged to present a visual figure of the text so that the text is not so arduous.

d. Line 266: 7 instances "among these participants". Please specify how many participants

e. It would be preferable if the authors walked the reader through something like figure 6 illustrating the patterns discussed in this paragraph, rather than concluding with a pointer to a figure that the readers must make sense of on their own. This reviewer believes that representations like figure 6 should be more common in this manuscript so that the patterns of transmission and translocation can be deduced across subjects. This would make the data actionable in a way that summary statistics just don't.

12. In the discussion, the authors mention "We found that participant hands transmitted both MRSA and VRE, but not R-GNB, to environmental surfaces outside of their rooms." This raises the unanswered question as to whether these species were co-transmitted with one another or whether these transmission events occurred independently among different individuals. Again, this highlights the need for the results to be presented on a per-subject manner.

13. The authors note that patterns were studied across 3 NHs but do not mention whether patterns of transmission differed across facilities in the main text. This seems like something worth at least a supplementary figure

14. "New MDRO acquisition in the study population was balanced by spontaneous loss of MDROs. Overall, 24 of 117 (20.5%) participants not colonized with an MDRO on enrollment later acquired and were discharged with a new MDRO

a. The authors should perhaps break this down by MDRO, rather than presenting the data in the aggregate. Do we see differences in the rates of acquisition or loss by pathogen? Do we see differences in the number of people who are acquired by all pathogens vs. 1 pathogen who are discharged with or without a new MDRO

Data reproducibility

b. The data and code availability section is missing

c. SNP calling – the authors are encouraged to report the specific commands used to do their bioinformatics analysis. The current method section is insufficient for another investigator to reproduce the findings

i. "sequencing, of which 414 isolates from 26 patients passed QC": please specify the QC protocol used and the method used to confirm species identification (16s gene? ANI?)

ii. Why were the reference genomes for bacterial read mapping chosen? Had you chosen a strain from your outbreak would we have seen more core genome SNPs? How many reads mapped to these alignments and what fraction of the genome did the total alignment cover?

iii. "variants were called and filtered using Samtools v.1.11" please specify the commands used to call and filter the SNPs.

iv. Only variants located at nucleotide positions present in all isolates were considered for the transmission analyses: does this mean that positions with Ns were eliminated? What % of positions had Ns?

v. How was mlst defined?

d. All the tools used for bioinformatics analysis should be reported in the reporting summary along with their version numbers

e. The reporting summary does not describe the full experimental design including the numbers of individuals at each of the three nursing homes

f. "We followed enrolled participants for up to three months or until discharge, whichever came first." Please specify the interval of sample collection.

15. Discussion

a. The higher proportion of VRE and R-GNB colonization at baseline and acquisition while in the VA NH may be due to differences in the inherent transmissibility of these organisms or may reflect the long-term effects of the enterprise-wide MRSA surveillance policy implemented by the VA. <<< could this also be explained by the samples that you took? Would we expect to see different patterns if we sampled different body sites?

b. It was overall very difficult for me to evaluate the discussion given the considerations I have outlined above.

Minor points

1. Using newer genomic methods: standard genome sequencing isn't exactly new....I suggest just saying using genomics

Version 1:

Reviewer comments:

Reviewer #1

(Remarks to the Author)

The authors have thoroughly and expertly addressed all feedback that I provided to them. This paper provides noteworthy results based on robust methods that should be published.

Emily Sickbert-Bennett

Reviewer #2

(Remarks to the Author)

Thank you for the robust revisions and carefully addressing the comments. The added detail and explanations, in addition to the revised and novel figures, make this a clear and informative manuscript.

Reviewer #3

(Remarks to the Author)

The manuscript epidemiology of multidrug resistant organisms across three VA nursing homes and MDRO transmission during interactive visits by Mody et al. is much improved. The authors addressed the bulk of my concerns, and in its current form I believe the manuscript will be a valuable addition to the literature. I still take issue with the claim that they are observing spontaneous decolonization. It would be clearer to relabel 'spontaneously decolonized' as 'not detected' in figure 2 and elsewhere where that term is used. The term 'decolonized' implies a definitive outcome, but I'm concerned that what's described as 'spontaneously decolonized' could be a result of reaching the limit of detection (LOD) in addition to decolonization. To establish decolonization, a formal limit of detection analysis should be performed. This is the only way that a claim of decolonization can be rigorously supported. The unknown transmission event in participant 2, for example, could be an instance of this strain falling below the limit of detection on May 3, May 5 and May 10 on the hands. Throughout the manuscript, it is the opinion of this reviewer that the authors should soften language arguing that they are observing decolonization by using "not detected" if a LOD analysis is not performed.

Minor points:

1. Figure 1

- a. In the figure legend for figure 1, state the statistical test used
- b. Please state N for Facility A, B, C in the figure legend

2. Supplementary figure 2

- a. Please state how the combinations are plotted. I would've expected that Hands and Nares would represent a composite of both the hands and nares. Rather, it looks like maybe only people who are colonized by the same species at the hands and nares is shown. Please clarify in the figure legend

In response to Reviewer comments below, we now have 23 tables and figures in total, of which many formerly-submitted tables or figures are referenced in a new place in the text. Former and new numbers for all tables and figures in the main text and supplement are found below:

Main Paper -- 5 tables; 5 figures

*We removed former Figure 3 altogether, per Reviewer 1, comment 10.

Former Table/Figure	NEW Table/Figure	Title
Table 1	Table 1	Baseline characteristics of study participants, by facility
Supplemental Figure 4 (partially)	Table 2	Participant colonization and environmental surface contamination at study baseline (during the first in-room visit)
Table 2	Table 3	Association between participant characteristics and baseline MDRO colonization or new MDRO acquisitions
Supplemental Figure 4 (partially)	Table 4	New acquisition of MDROs on participant and in participant rooms during the study period
Supplemental Table 5 (partially)	Table 5	Transmission prevalence among various interactive visits attended
Figure 2	Figure 1	Colonization with multidrug-resistant organisms at baseline, at discharge, and at any time during the study
Figure 4	Figure 2	Changes in MDRO colonization status from baseline visit to discharge (last study visit)
Figure 5	Figure 3	Transmission of DNA-confirmed MRSA or VRE to participant hands or equipment
Figure 6	Figure 4	Genomics support of microbiological linkages among two participants
Figure 1	Figure 5	A representative interactive visit that resulted in a transmission event

Supplement -- 6 tables; 7 figures

Former Table/Figure	NEW Table/Figure	Title
Supplemental Table 2	Supplemental Table 1	Significant characteristics of Veterans Affairs nursing home participants, by colonization status at baseline
Supplemental Table 3	Supplemental Table 2	Significant characteristics of Veterans Affairs nursing home participants, by acquisition of new MDROs
Supplemental Table 1	Supplemental Table 3	Categorization of participants for sequencing selection, based on MDRO colonization, room contamination, new acquisition, and transmission
Supplemental Table 4	Supplemental Table 4	Veterans Affairs nursing home characteristics and infection control policies
	Supplemental Table 5 (NEW)	Number of swabs positive out of swabs collected during in-room visits
	Supplemental Table 6 (NEW)	Number of swabs positive out of swabs collected during interactive visits
Supplemental Figure 2	Supplemental Figure 1	Resident enrollment

	Supplemental Figure 2 (NEW)	Participant body sites colonized during in-room visits at facility A (a), B (b), and C (c)
	Supplemental Figure 3 (NEW)	Participant body sites colonized during interactive visits at facility A (a), B (b), and C (c)
	Supplemental Figure 4 (NEW)	Changes in MDRO colonization status from baseline to discharge at facilities A (a), B (b), and C (c)
	Supplemental Figure 5 (NEW)	Genomics support of microbiological linkages among two participants
Supplemental Figure 1	Supplemental Figure 6	MRSA and VRE strain diversity in the overall participant population and between epidemiologically linked carriers
	Supplemental Figure 7 (NEW)	Phylogenies for: a. Facility A E. faecium ; b. Facility A S. aureus ; c. Facility B E. faecium ; d. Facility B S. aureus ; e. Facility C S. aureus ; and f. All E. faecalis

*We removed former Supplemental Figure 3 altogether, per Reviewer 1, comment 10.

Reviewer #1 (Remarks to the Author):

General Comments:

This prospective cohort evaluation of MDRO transmission at 3 nursing homes is a unique addition to the literature as the investigators are able to describe and measure the transmission dynamics within the nursing home setting between patient/surfaces/staff. The whole genome sequencing analysis adds another fascinating layer of complexity to the review of transmission patterns showing how multiple strains can be involved in acquisition. The authors have presented the rationale and methods in a very clear manner. The overall results are clearly stated, however, some of the specifics within the tables/figures could be even more impactful with some clarification and even by condensing the number of tables/figures (see below for specific suggestions).

Response: *We appreciate the positive comments above.*

Specific Comments:

Abstract

1. Lines 49-50, we would recommend adding details to include environmental sampling of surfaces and hands as these represent a great deal of work and the key findings of the paper.

Response: *We agree. The following sentence has been added to the abstract (lines 30-33): "Participant hands, nares, groin, and seven environmental surfaces were swabbed during 758 regularly scheduled in-room visits; participant hands, healthcare personnel hands, and equipment were swabbed during 345 unscheduled interactive visits."*

Methods

2. Lines 109... , please include a statement about whether disinfection protocols or access to hand hygiene were modified as a result of participation and knowledge about this study.

Response: *Thank you for this important point. The following sentence has been added to the methods (which is now the last section of the paper, line 339): "No disinfection policies nor hand hygiene access were modified in any way because of participation in this study." During interactive visits, healthcare providers were encouraged by study staff to perform their care activities as they typically would, e.g., with no alterations due to the study team presence.*

3. Lines 117-118, please include details about what surface area of the hands were swabbed.

Response: *The size of participant hands vary, but we always swab the entire palm of both hands and the tips of each finger all participants, this has been added to the Microbiologic Methods (lines 340-341).*

4. Line 120, please comment on any known information about the sensitivity of the environmental and hand sampling. This information is needed for better interpretation of the acquisition/loss findings and should be included in a short statement in the discussion.

Response: *We thank the reviewer for this comment. Our approach to hand and environmental surface sampling, during both in-room and interactive visits, is to enrich each swab overnight in BHI broth, which confers high levels of sensitivity. As a result, we believe that contamination negatives are in fact true negatives. We have used this approach in a few studies (Mody L, et al. Longitudinal assessment of multidrug-resistant organisms in newly admitted nursing facility patients: Implications for an evolving population. Clin Infect Dis 2018;67(6):837-844; and Cassone M, et al. Environmental panels as a proxy for nursing facility patients with methicillin-resistant Staphylococcus aureus and Vancomycin-resistant Enterococcus colonization. Clin Infect Dis 2018;67(6):861-868). We have added more specific details to our Microbiologic*

Methods section (lines 351-362) and have added limitations in the discussion (lines 262-266 & 274-276).

Results

5. Please reconcile the number of participants included between Supplemental Tables 2-3 and Table 1 and with the enrollment numbers in Supplemental Figure 2. Simple footnotes about these differences at the bottom of Supplemental Tables 2-3 would be helpful.

Response: *We thank the reviewer for pointing this out. Data included in these tables excluded a small number of participants who had partial missing data. Footnotes have been added to clarify this in Supplemental Tables 1 and 2 (formerly Supplemental Tables 2 and 3), and details in the Methods have been added (lines 455-460). We have stratified these tables by the 3 facilities (per Reviewer 3's comment 1) and have included only significant variables in Supplemental Tables 1 & 2; however, we are happy to add in all variables if the Editor prefers.*

6. We find Figure 4 to be more informative and a better representation of the main findings of the study compared to Figure 2. You may consider replacing Figure 2 with Figure 4.

Response: *We agree that former Figure 4 (now Figure 2) is very informative and illustrates significant findings. Due to our expanded analyses per Reviewer 3's comments (i.e., stratifying all results by facility/cohort), we have kept former Figure 2 (now Figure 1) in the main paper; however, we are happy to move to the Supplement if the Editor prefers.*

7. Line 224-225, the higher ADL scores are not overwhelming based on the magnitude of the OR and the 95% CI overlapping the null value. You may consider not highlighting that prominently in the text or providing more rationale for highlighting this finding. This point is also referred to in line 281.

Response: *The reviewer makes a good point. We have removed ADL score as a significant risk factor in the results (line 108-110) and added text in the discussion (line 231), clarifying ADL scores was not a statistically-significant risk factor for MDRO colonization in our study.*

8. Figure 5, even with careful reading of the legend and axes, we think this figure is difficult to interpret as currently presented. The readers will be quick to think these % represent how often equipment was not contaminated vs. contaminated rather than % of transmission. This figure is so central to the main unique findings of this paper it may be worth exploring if there is a clearer way to present it.

Response: *We agree with this reviewer that former Figure 5 has much room for improvement. A new version of this figure, now Figure 3, is now included in the paper.*

9. Line 256, please specify which categories from Supplement Table 1 were used to select the 14 participants for sequencing.

Response: *We have added to the Results (line 193) that participants within category 4b and 4c of former Supplemental Table 1 (now Supplemental Table 3) were focused on for genomic analysis.*

10. If space for figures is limited, Figures 3 and supplemental Figures 3 may not be needed beyond including the main findings directly in the text.

Response: *We agree with this reviewer and have removed former Figure 3 and Supplemental Figure 3 from the paper and supplement.*

Discussion

11. Lines 279-282 mentions risk factors that Table 2 did not show to be statistically significant

findings. Please delineate between your study's findings and those that you are referencing in the literature.

Response: *We thank the reviewer for pointing this out. We have removed indwelling device use from this sentence since we did not find an association between device use and MDRO colonization in our study, though other studies have (line 231).*

12. Line 323, Please consider also mentioning whether you believe your findings to be generalizable to non-VA nursing home patients.

Response: *The reviewer brings up an excellent point. VA vs. non-VA nursing home populations often vary in their gender distribution and in the types of interactive visits they offer on-site—i.e., VA populations are predominantly male and offer services beyond PT and OT, including dialysis, ophthalmology, radiology, radiation, etc. We have added that these data may not be generalizable to non-VA NHs in the Discussion (line 282); however, Gontjes et al. similarly looked at transmission events during PT/OT sessions at non-VA NHs and found rates of transmission very similar to this current analysis, as we include in the discussion (lines 251-255). Further studies are needed comparing VA and non-VA NH populations to see how MDRO transmission might vary in each setting.*

Reviewer #2 (Remarks to the Author):

This important work by Mody and colleagues is a welcome addition to our understanding of colonization, transmissions, and infections with drug-resistant bacterial pathogens in the NH setting. The focus on interactive visits paired with patient and room environment baseline of colonization is a noteworthy and underscores the potential of MDRO transmission during encounters outside the resident rooms. This was a multicenter cohort study of 3 VA NHs with nearly 200 participants included, providing ample data and analysis opportunities using complementary methods to address the study's primary objectives of quantifying MDRO prevalence, transmission events, and transmission dynamics. This is a novel study that represents a considerable effort to characterize carriage rates of important MDROs as well as their transmission within a complex and relatively understudied subpopulation of NH residents. I recommend this for publication following the clarification of these points:

Response: *We thank the reviewer for their praise and recommendation for publication.*

The authors present detailed epidemiological data to make inferences about putative transmission events, however it is unclear if the WGS data was fully used to support the microbiologic tests representing likely transmission or those transmissions that did not have any detected sources. Considering the effort of sequencing 338 isolates, why are there no species-specific whole-genome phylogenies or contact network analyses presented? In Figure 6, five *E. faecium* strains are presented in the two examples of putative transmission, yet phylogenetic and/or network context is missing to explain strain identification from the much larger pool of isolates and transmission events that are not presented.

Response: *We thank the reviewer for this pertinent request for more genomic information. We have added whole-genome phylogenies for all three resistant organisms to the supplementary materials (Supplementary Figure 7). Examining these phylogenies one can see the dominant signal of clustering of isolates from participants and their in-room and out-of-room environments. These provide strong support for residents contaminating their surroundings. There are also a few instances where you can see samples associated with different participants inter-mixed, which is a signature of between-resident transmission. While our focus on dense sampling of individual residents, versus more limited sampling of a more comprehensive set of residents, precludes meaningful studies of resident-to-resident transmission, these phylogenies do*

demonstrate the power of capturing within resident diversity for detection of transmission events.

Minor comments:

It is unclear how the species-specific SNV thresholds were calculated and what percentage of within-patient diversity these would cover? The cited reference to support lines 145-147 refers to a study of *C. difficile* where a threshold of 2 SNVs was implemented to identify recent transmissions; however the implementation of 10 and 12 SNVs for VRE and MRSA, respectively (lines 171-173), are different and based on data presented in Supplemental Figure 1, although these values are not sufficiently explained.

Response: *We apologize for the insufficient detail provided regarding our selection of SNV thresholds. We referenced the C. difficile manuscript as prior work by our team members using intra-patient diversity to establish organism-specific and context-specific thresholds. Employing a similar approach for VRE and MRSA we identified thresholds of 10 and 20 SNVs respectively (Supplementary Figure 6), below which 98% of pairs are intra-residents for VRE and 95% for MRSA. Thus, based on this analysis we expect these cutoffs to balance sensitivity and specificity in identifying participants/healthcare personnel/equipment plausibly linked by direct transmission. Note that these cutoffs are also consistent with prior studies establishing thresholds based on epidemiologic data for these organisms (Coll F, et al. Definition of a genetic relatedness cutoff to exclude recent transmission of methicillin-resistant Staphylococcus aureus: A genomic epidemiology analysis. Lancet Microbe 2020;1(8): e328-e335; and Gouliouris T, et al. Quantifying acquisition and transmission of Enterococcus faecium using genomic surveillance. Nat Microbiol 2020;6(1):103-111). Further detail on SNV thresholds has been added to our Methods (lines 392-401).*

Figure 4: Decolonization implies active measures of IPC to remove target organisms – perhaps revise to “spontaneous decolonization” or “not detected”?

Response: *This is an excellent point. We have changed the term, “decolonized,” to “spontaneously decolonized” in Figure 2 (former Figure 4) and maintained “spontaneous loss of colonization” in our text.*

Reviewer #3 (Remarks to the Author):

Mody et al. presented the manuscript, EPIDEMIOLOGY OF MULTIDRUG-RESISTANT ORGANISMS (MDROS) ACROSS THREE VA NURSING HOMES AND MDRO TRANSMISSION DURING INTERACTIVE VISITS". The authors conducted a multisite, prospective cohort study at three VA nursing homes and used genomics to examine colonization and potential transmission of MRSA, VRE, and R-GNB during activities at the nursing homes in a subset of 27 individuals. A major strength of the manuscript is the study design which examined 3 facilities. My first major concern with this manuscript is that it is currently very difficult to read – 1) it is written as if it were a clinical trial, but according to the reporting summary it is not and 2) the current display items are of interest to a very narrow audience interested in infection prevention. The strengths of the data were not adequately leveraged to answer questions about the extent to which we see differences in the pattern of transmission of MRSA, VRE or other pathogens across the 3 facilities or the types of interactive visits. Since patients were treated as a single population across the 3 facilities, the analysis is not reflective of the multi-site cohort design. Technically, I have concerns about the way colonization and transmission are defined. Likewise, the reader is unable to discern whether "transmission" frequencies appear to be greater for one species relative to others or with respect to the time interval separating sample collection (1 day vs. 1 or 3 months). Moreover, it is unclear as to whether we see multiple body sites of a single individual colonized by one pathogen or how often we see multiple pathogens colonizing a single individual. It is the opinion of this reviewer that for this manuscript to be of broad interest to people outside of infection control a major reworking of the analysis and the main display figures is essential. Since this is not a specialized journal focused on infection prevention or clinical trials, many of the current display items would be better suited to the supplement so that main figures related to the primary objectives – and the microbiology/genomic epidemiology -- can be included. The current analysis limits my enthusiasm for the manuscript. Finally, the methods are not sufficiently written – I would not be able to reproduce the genomic findings, for example, even given the data in SRA, and the reporting summary should be more comprehensively completed.

Response: *We thank the reviewer for their thoughtful, thorough review. All of the above concerns have been addressed in their individual comments below.*

Major points

1. "We conducted a multisite, prospective cohort study at three VA NHs (Ann Arbor, MI, Detroit, MI, and Cleveland, OH) from April 2021 to September 2023." It's important to note that in the main text and in the reporting summary the 3 cohorts are treated for the most part as if they belong to a single population. If the investigators wish to position this paper as a multisite prospective cohort study, then it's important to assess the magnitude of differences in the outcome variables based on the facility in which each cohort was sampled. For this work to be considered of general interest to readers outside of infection prevention community the authors may want to consider reworking the paper to explore the cohort angle a bit.

Response: *Generally, multisite cohort studies help with generalizability of findings when compared with single-site studies. Furthermore, all these three sites were VA sites and recruited only Veterans, but we do acknowledge the reviewer's comment. In response, all previously reported tables/figures, as well as all new tables and figures added as a result of reviewer comments, have been stratified by facility.*

1a. The reader has to look at supplementary figure 4 to see the distinct environmental sites that were sampled. Given that differences in infection prevention practices (not just policies) could

lead to different transmission dynamics it is suggested that these data for participant room in sFig4 be presented on a per-facility basis

Response: *We thank the reviewer for this comment. Supplemental Figure 4 has been changed to a table, now stratified by facility, and moved to the main paper (Tables 2 and 4). If the Editor prefers to see this data in figure form, we are happy to bring back former Supplemental Figure 4 into the main body, since there were no major facility-level differences. Hence, we would not be able to study the role of facility-level infection prevention practices in our outcomes.*

1b. It seems to me that a main finding of interest is whether the number of transmission events varied based on the type of interactive event, the time interval separating interactive events, and the facility in which the participant was located. I don't understand why this isn't presented as a main figure. Is it that regardless of interactive visit the feature determining transmission was hand colonization? For example, of the 17 visits where there was equipment contamination with MRSA and transmission, were they all different kinds of visits? Do we see higher frequency within vs. out of room?

Response: *The reviewer is correct that transmission events during interactive visits is a novel aspect of our study and merits further emphasis in the main paper. New Table 5 in the main paper, an expansion of former Supplemental Table 5, now shows the number of each type of interactive visit attended at each facility (denominator) and the number of visits that included at least one transmission event (numerator). Also included is the directionality of these transmission events—i.e., whether the transmission of an MDRO occurred to a surface/equipment following interaction, or to a participant's hand following interaction. Across all three facilities, 15.9% of all interactive visits involved the transmission of an MDRO (either MRSA, VRE, or RGNB), more commonly to a surface (10.1% of visits) compared to transmission to the participant hand (6.6% of visits). This table is indeed very busy. If the Editor wants us to present aggregate data for simplicity, we are happy to do so. Our former Figure 5 (now Figure 3) has also been revamped to more clearly illustrate our key transmission findings among MRSA and VRE transmission events. These interactive visits were dependent on resident appointments and were not predetermined.*

2. Since we are talking about transmission and de-colonization events, the summary statistics about sample numbers is an insufficient way to describe the study design. It's important for the reader to know whether we're looking at multiple colonized body sites of one person, whether multiple species were found on the same person, as well as the number of time points over which we observe an apparent "colonization" or transmission event per person. The authors are encouraged to present a figure that displays subject, body site, and time point with information about the type of interactive visit and to present this information for each facility. Since it appears figures were made in R, this could be accomplished with some clever use of the facet grid function and the shape/color function of geom_point. This reviewer believes this is essential in the reformulation of this manuscript and necessary in the supplement.

Response: *We thank the reviewer for this suggestion. We have added Supplemental Figures 2 and 3, which show participant body site colonization with MRSA (in blue), VRE (in red), and RGNB (in green) at each in-room visit (Supp F2) and interactive visit (Optional Supp F3) conducted during the study period. Each participant is represented by a single dot, and all three facilities are displayed to show the differences between them. We're happy to remove Supplemental Figure 3 based on the Editor's preference.*

It's worth noting here, as we clarify in several subsequent responses, that in-room visits were conducted at predetermined intervals—at study baseline (day 0), and then day 7, 14, 21, 30, 60, and 90—whereas interactive visits were conducted based on a participant's appointment to be seen or evaluated at locations such as physical therapy, occupational therapy, etc. The appointment times were set by the services.

2a. Presentation of cohort design. The subjects who meet certain criteria for which transmission/de-colonization will be studied should be presented as well as an overview of the types of visits (frequency of each kind) should be presented.

Response: *We thank the reviewer for this comment. We have clarified the two types of visits conducted in the study and their frequency in the Methods (lines 309-315). We collected in-room samples at study baseline and follow-up visits on a **weekly basis** for the first month, followed by monthly thereafter (i.e., days 0, 7, 14, 21, 30, 60, 90). In addition to these regularly-scheduled in-room samples, we attended a maximum of 5 interactive visits with each participant, such as PT, OT, dialysis, or dining room visits. The interactive visits were not conducted at a predetermined frequency (like the in-room visits), because every participant's need and capability for services, as well as schedule, is different. Interactive visits were accomplished based on participant appointment times. Additionally, we have clarified our definitions of colonization, contamination, new acquisition, and transmission in the Methods and specified all analytic sample Ns in the "Main outcomes and definitions" section.*

2ai. It is important for the reader to understand how many samples of which variety the 185 participants contributed over time – as well as the number of individuals in each facility. This is to help the reader understand whether a more targeted analysis of a subset of subjects would better serve the questions posed.

Response: *We agree with the reviewer. A table has been added to the Supplement (Supplemental Table 5) which shows the total number of samples from all sites collected during in-room visits, as well as the number of samples positive for MDROs at any time.*

2aii. Likewise, the reader needs to understand how many samples of which variety the 134 participants contributed during the interactive visits.

Response: *We agree with the reviewer. A table has been added to the Supplement (Supplemental Table 6) which shows the total number of samples from all sites collected during interactive visits, as well as the number of samples positive for MDROs at any time.*

3. Furthermore, since samples were collected on a daily basis for a month and on a monthly basis thereafter it's important for the analysis to report whether we different dynamics based on the interval separating sample collection. Do we see putative transmission events more often when we look at samples separated in time by 1 day vs. 2 or 7 days, and are we calling events transmission events when they are separated by more than 1 month? If transmissions are linked by a month or more, then describing the event as a transmission seems like it could be a bit of a stretch.

Response: *The reviewer brings up a good question. Transmissions were not assessed nor defined relative to time; rather, a transmission was a single event within an interactive visit, thus documenting transmission in real time (see Figure 5).*

To clarify, samples were not collected on a daily basis. We have clarified the two types of visits conducted in the study and their frequency in the Methods (See our response to Reviewer 3, comment 2a, lines 309-315). Our definition of transmission has also been clarified (lines 429-432): "Transmission was defined as a change in microbial colonization or contamination from negative to positive following use or interaction, when swabs were collected *both* before and after the interaction, within a single interactive visit. Transmission events were only assessed during interactive visits (not in-room visits), as swabs could feasibly be collected *both* before use/interaction and after use/interaction." We believe these changes clarify our analytic strategy.

4. Were any of these individual roommates and how did sharing a room impact transmission frequencies.

Response: *The goals of this paper were to look at transmission during out-of-room interactive visits. Examining transmission events between roommates was beyond the scope of this study; however, our group has examined MDRO prevalence among roommates in a prior study (Cassone M, et al. Not too close! Impact of roommate status on MRSA and VRE colonization and contamination in nursing homes. Antimicrob Resist Infect Control 2021;10(1):104).*

In response to the reviewer's comment, we did look at the number of participants sharing a room at any given time. Private rooms were much more prevalent during the study period, owing to COVID-19 restrictions in place at all participating sites. However, there were three pairs of participants who were roommates for at least one in-room visit during the study period, while 5 pairs of participants had their own room but shared a bathroom for at least one in-room visit during the study. These numbers were too few to make any meaningful conclusions.

5. Prevalence of colonization

5a. First, how is colonization defined, given that multiple body sites were sampled? Please specify on line 216.

Response: *We thank the reviewer for this comment. The following text has been added to the Methods for clarity (lines 407-412): "For outcome one, a participant was considered colonized at study baseline if an MDRO (MRSA, VRE or R-GNB) was detected at any one of the three participant body sites sampled (i.e., nares, groin, or hand). Similarly, if an MDRO (MRSA, VRE or R-GNB) was detected at any one of the seven environmental surfaces sampled (i.e., bed controls, bed rail, call button, television (TV) remote, bedside tabletop, privacy curtain, or toilet seat), the participant's environment was considered contaminated with that MDRO. We use "colonization" to refer to the participant, and "contamination" to refer to the environment"*

5b. Second, with respect to defining colonization, how do we distinguish between contamination of a surface with an MDRO and colonization, which implies stable growth? Do you have instances in which you observe a "transmission" and subsequent evidence of growth at that same site?

Response: *We use "colonization" when referring to MDROs detected at any one of the three participant body sites; we use "contamination" when referring to MDROs detected at any one of the seven environmental sites sampled. We have clarified this in the Methods (lines 411-412). We do see some persistent growth of MDROs on both surfaces and participant body sites during in-room visits (collected weekly for one month and monthly thereafter); we will conduct a more in-depth analysis in a follow-up paper to explore persistent vs. intermittent carriage on participants and on surfaces.*

Among the 55 interactive visits where we observed at least one transmission event, 23 visits had a transmission to a participant hand (i.e., the participant's hand started off clean but became colonized after touching surface(s)). Following eight of these 23 interactive visits (35%) with a transmission to the participant hand, there was either an in-room or a new interactive visit where the participant's hand continued to be colonized with the same organism, which suggests persistency following transmission.

5c. Third, how do these patterns differ across the nursing home settings? If the collection of samples across facilities is important, then the data should be analyzed by facility, rather than aggregated as if the samples were all drawn from a single facility.

Response: *In general, for epidemiologic investigations and for the purpose of this project, it is important to recruit residents from various facilities. This also enhances generalizability. Our power calculations described in our externally-reviewed grant were based on the premise that we will analyze all data in aggregate. However, based on the reviewer's comments, we have*

included all results stratified by facility (Tables 1, 2, 4, & 5; Figures 1 and 2 (2 is stratified in Supplemental Figure 4); Supplemental Tables 1, 2, 4, 5, & 6; Supplemental Figures 1, 2, 3, & 4).

5d. Finally, what is the limit of detection for each of the species that you're surveying? If you were to sequence samples, would you identify *S. aureus* (and other species) reads and/or the genes conferring abx resistance? Given that we're trying to understand transmission or strain "displacement", it's very important for the reader to believe that the colonization negatives are in fact true negatives.

Response: *Please see our response to Reviewer #1, comment #4. Our approach to hand and environmental surface sampling, during both in-room and interactive visits, is to enrich each swab overnight in BHI broth, which confers very high levels of sensitivity, theoretically to a limit of 10 individual cells per sample, while avoiding false positives resulting from DNA remnants of dead bacteria. As a result, we believe that contamination negatives are in fact true negatives. Once we receive swabs into our lab, we identify individual organisms using traditional, culture-based, microbiology techniques. We test and identify phenotypically unique colonies suggestive of *S. aureus*, for example. Strains from the interactive visits that indicate a transmission event are sequenced further to establish similarities. We have added more specific details to our Microbiologic Methods section (lines 351-362) and have added limitations in the discussion (lines 262-266 & 274-276). In the future, it might indeed be possible that we can sequence pathogens from clinical samples directly; however, for the purposes of this large, novel population study, we pursued published and accepted methods. It will be our future goal to test direct sequencing in a smaller sample.*

5e. With respect to colonization positivity/negativity, the reader needs to have a feel for the numbers of colonies for each species that was plated and sampled and identified. Otherwise, it is hard to understand whether had you sampled more colonies you'd see colonization at different time points. The number of colonies sampled for each species should be presented. If you were to do a rarefaction curve, would it be saturated?

Response: *Details on microbiology techniques, including colony selection and plating, are now included in lines 351-362. All colonies that appeared different from others on the plate were tested, unless clearly inconsistent with the expected appearance of each target organism. For example, if three different colonies were seen, each potentially consistent with *S. aureus*, they were all tested separately. This minimizes the risk of inadvertently missing a target species organism from a sample. Although rarefaction curves may not provide a perfect indication in our case due to uneven distribution of the number of sampled colony per sample, we expect that it would reach a horizontal slope almost immediately, species-wise. However, it is true that within a species, different strains could be present at the same time, some of which could be missed because of phenotypically indistinguishable colonies. This has now been added to the limitations section (lines 262-266 & 274-276). Although we do not expect that different strains would be present in a sample very often, this could certainly be the case in some instances. Although not a main focus of the present paper, this information is very useful epidemiologically and a motivation to set up further studies.*

5f. Finally, how would these patterns change were you to have sampled all the isolates of a given species, not just the ones that were antibiotic resistant? It could be that the selection method gives us an under-estimate of colonization positivity (e.g., someone is colonized with *S. aureus* which acquires abx resistance). For this reason, it's important for the authors to spell out the methods used for selection of these species, rather than referring the reader to an external set of sources (lines 122-124). Certainly, there should be a limitations section that acknowledges that the study design may have led to an incomplete assessment of the prevalence of colonization as well as the appearance of acquisition events.

Response: We agree with the reviewer that it would be interesting to understand transmission of antibiotic-sensitive organisms, e.g. MSSA, that would have the potential to acquire antibiotic resistance. Our interest in this population was to understand transmission of drug-resistant organisms first. We now provide further details in our microbiology methods (lines 351-362) and have added a limitation to the discussion section (lines 276-280).

5g. the authors should consider presenting an analysis of the pan genome relative to the core genome, rather than relying simply on SNPs, particularly since some of these SNP distributions are extraordinarily small. 20 SNPs isn't much genomic variation. Similarly, while the authors mention sequence typing was performed, I don't see these data presented anywhere

Response: We thank the reviewer for their suggestion. While expanding to the pan-genome may increase resolution, it also adds complexity in potentially adding variation that is not acquired by vertical descent, which violates assumptions being made when using genetic distances to infer the timing of putative transmission events. Moreover, examining the distribution of intra- versus inter-resident SNVs (Supplemental Figure 6), as well as whole-genome phylogenies (Supplemental Figure 7), supports the resolution provided by a core-genome approach to distinguish between strains circulating in the facility.

6. "Of these, four participants were colonized with all three MDROs at enrollment and thus not considered at risk of acquiring any new MDRO, resulting in 181 participants with 725 in-room follow-up visits." This reviewer would appreciate an analysis of the clones at baseline and at follow-up for the individuals who were colonized at enrollment. Did we see the same strain at follow up or do you believe there was a transmission and displacement event at follow-up? Do we see episodes of mixed colonization where two strains are present? Importantly, the time scale of sample collection is also important...

Response: The reviewer brings up a good point—the four participants colonized with all three MDROs (MRSA, VRE, and R-GNB) at baseline have interesting stories and heavy colonization that merits it's own publication. Further detail regarding the four participants is included below. Whole genome sequencing has been completed on MRSA or VRE isolates from two of these four participants. For those two with sequencing complete, there is persistency of the VRE and MRSA strains present at baseline as well as simultaneous intermittent carriage of additional VRE and MRSA strains.

- **Participant 1**
 - We conducted six in-room visits (0,7,14,21,30,60, between 5/4 and 6/30) and three interactive visits with this participant (OT on 5/4, dialysis on 5/5, and PT on 5/6). See details of each visit below:

Date of visit	Type of visit	Participant colonization	Room contamination	Transmissions
5/4/21	In-room (0)	Nares: MRSA Groin: VRE & R-GNB	Toilet seat: VRE	-
5/4/21	Interactive visit #1 (OT)	-	-	None
5/5/21	Interactive visit #2 (Dialysis)	-	-	1 MRSA transmission: unknown source → resident hand
5/6/21	Interactive visit #3 (PT)	-	-	None
5/11/21	In-room (7)	Nares: MRSA	Toilet seat: VRE	
5/18/21	In-room (14)	Nares: MRSA Groin: VRE	Toilet seat: VRE	

5/25/21	In-room (21)	Nares: MRSA	Toilet seat: VRE TV remote: MRSA	
6/1/24	In-room (30)	Nares: VRE	Toilet seat: VRE	
6/30	In-room (60)	None	Toilet seat: MRSA	

- **Participant 2**

- We conducted six in-room visits (0,7,14,21,30,90, between 6/23 and 9/21) and four interactive visits with this participant (PT on 6/23, PT on 6/30, PT on 7/14, and Dialysis on 9/21). See details of each visit below:

Date of visit	Type of visit	Participant colonization	Room contamination	Transmissions
6/23/21	In-room (0)	Nares: MRSA Groin: VRE & R-GNB	Curtain: MRSA	-
6/23/21	Interactive visit #1 (PT)	-	-	2 VRE transmissions: unknown source→surface (gait belt); unknown source→HCW hand
6/30/21	In-room (7)	Nares: MRSA Groin: VRE	Table top: MRSA	
6/30/21	Interactive visit #2 (PT)	-	-	1 VRE transmission: unknown source→surface (blood pressure cuff)
7/7/21	In-room (14)	Nares: MRSA, VRE Groin: VRE, R-GNB Hand: MRSA	Table top: VRE TV remote: MRSA, VRE Bedrail: MRSA, VRE	-
7/14/21	In-room (21)	Nares: MRSA, R-GNB Groin: VRE, R-GNB	Table top: VRE TV remote: MRSA, VRE Curtain: VRE Bedrail: VRE Toilet seat: VRE	-
7/14/21	Interactive visit #3 (PT)	-	-	None
7/21/21	In-room (30)	Groin: VRE Hand: VRE	Bed control: MRSA Table top: MRSA Bedrail: MRSA	-
	In-room (60) - Skipped			
9/21/21	In-room (90)	Nares: MRSA Groin: MRSA Hand: MRSA	Bed control, Call button, Table top, TV remote, Curtain, Bedrail, Toilet seat: MRSA	-
9/21/21	Interactive visit #4 (Dialysis)	-	-	None

- **Participant 3**

- We conducted six in-room visits (0,7,14,21,30,90, between 6/23 and 9/21) and four interactive visits with this participant (PT on 6/23, PT on 6/30, PT on 7/14, and Dialysis on 9/21). See details of each visit below:

Date of visit	Type of visit	Participant colonization	Room contamination	Transmissions
9/21/21	In-room (0)	Nares: VRE Groin: VRE & R-GNB Hand: MRSA, VRE	Bed control: MRSA Call button: MRSA, VRE Table top: VRE TV remote: VRE Curtain: VRE Toilet seat: VRE	-
10/6/21	In-room (7)	Nares: VRE Groin: MRSA, VRE, R-GNB Hand: MRSA, VRE	Bed control: MRSA, VRE Call button: MRSA, VRE Table top: MRSA, VRE TV remote: MRSA, VRE Curtain: MRSA Bedrail: MRSA, VRE	-
10/13/21	In-room (14)	Nares: MRSA, VRE Groin: MRSA Hand: MRSA, VRE	Bed control: MRSA, VRE Call button: MRSA, VRE Table top: MRSA, VRE TV remote: MRSA, VRE Curtain: MRSA, VRE Bedrail: MRSA, VRE Toilet seat: MRSA, VRE	-
10/20/21	In-room (21)	Groin: MRSA, VRE Hand: MRSA, VRE	Bed control: MRSA, VRE Call button: MRSA, VRE Table top: MRSA, VRE TV remote: MRSA, VRE Curtain: MRSA Bedrail: MRSA, VRE	-
10/28/21	In-room (30)	Nares: VRE Groin: VRE Hand: MRSA, VRE	Bed control: MRSA, VRE Call button: VRE Table top: MRSA, VRE TV remote: VRE Bedrail: VRE	-

- **Participant 4**

Date of visit	Type of visit	Participant colonization	Room contamination	Transmissions
11/17/21	In-room (0)	Nares: R-GNB Groin: R-GNB Hand: MRSA, VRE	None	-
11/22/21	In-room (7)	None	TV remote: R-GNB Curtain: VRE Bedrail: R-GNB Toilet seat: R-GNB	-
12/1/21	In-room (14)	None	Curtain: VRE	-
	In-room (21) – Skipped			
12/16/21	In-room (30)	None	Curtain: VRE	-

7. Participants frequently acquired a new MDRO: MRSA (10.7% of 168 at-risk participants), VRE (29.7% of 138 at-risk), R-GNB (18.4% of 158 at-risk), and any MDRO (40.9% of 181 at-risk), with an average time to new acquisition being 14.7 days (range, 1-65 days). For the reader to understand the average time to new acquisition we need to have a better understanding of the number of days separating sample collection for each individual sampled. I'm also wondering whether the average time to acquisition differed by pathogen and/or the facility.

Response: *The frequency of sample collection has been clarified in our Methods (lines 309-315) and in multiple responses above (see Reviewer 3, comment 2a and 3). Since in-room samples were collected on days 0, 7, 14, 21, 30, 60, and 90, the number of days separating in-room samples was approximately 7, 7, 7, 7, 30 and 30 days, respectively. Interactive visit samples were not collected at pre-determined frequencies; the average time from baseline for interactive visits 1-5 was 9 days, 14 days, 20 days, 34 days, and 48 days (added to Supplemental Figure 3 legend). Average time to acquisition by pathogen is now included in the results (lines 140-142).*

8. With respect to figure 4, does this show colonization of any body site or colonization of a single body site? Please specify. If acquisition or loss at individual body sites per person is not shown, it should be in order for the reader to be convinced of acquisition or loss, particularly in light of concerns regarding the limit of detection.

Response: *Former Figure 4 (now Figure 2) shows colonization of any body site (nares, groin, or hand). Acquisition at individual body sites per person is now shown in Table 4.*

9. Again, with respect to supplementary table 5, it's important for the reader to understand the distribution of interactive visits across time for each subject and whether or not a transmission event occurred. A presentation of this type of raw data is necessary for the reader to interpret the summary statistics presented in the main manuscript.

Response: *We describe the average time between interactive visits in our response above (see Reviewer 3, comment 7) and have added the following to Supplemental Figure 3 legend— "On average, the first interactive visit occurred 9.3 days (SD 12.0) after study enrollment; the second interactive visit occurred 14.2 days (SD 14.7) after study enrollment; the third interactive visit occurred 20.3 days (SD 17.8) after study enrollment; the fourth interactive visit occurred 34.1 days (SD 26.0) after study enrollment; and the fifth interactive visit occurred 48.0 days (SD 31.2) after study enrollment." Interactive visits did not occur at a predetermined frequency, as we have clarified above and in the Methods (lines 309-315). Transmissions were not assessed nor defined relative to time; rather, a transmission was a single event within an interactive visit. Participant hand colonization across all interactive visits (1-5) is now included in Supplementary Figure 3, and Former Supplementary Table 5 has been expanded and is now in the main paper, Table 5. A substantial portion of our data are now part of our 23 tables and figures (10 in main body; 13 in supplement).*

10. "Equipment contamination with MRSA and VRE, however, was not associated with transmission to participant hands." How long after touching the surface was the hand sampled? Was there an attempt to see if over time you saw a different outcome?

Response: *As per our study protocol, study staff sampled the participant's hand immediately, in real-time, following contact with each individual environmental surface or equipment used during interactive visits. Study staff also sampled the participant's hand at the end of the interactive visit. Per our response to Reviewer 3's Comment 5b: 23 interactive visits included a transmission to a participant's hand (i.e., the participant hand started off clean but became colonized after touching surface(s)). Eight of these interactive visits with a transmission to the participant's hand (35%) were followed by either an in-room or a new interactive visit where the*

participant's hand continued to be colonized with the same organism; whereas 65% of these visits showed no follow-up colonization of the participant's hand at any subsequent study visits.

Potentially of interest to this reviewer is an observational pilot study our team conducted in 2019, sampling hospitalized patients known to be MRSA-colonized, multiple times over 90-minute intervals. The short-term recontamination rate within 90-minute intervals, even after disinfection at the beginning of each interval, was 27% for dominant hand of patients. (Wolfensberger A, et al. Understanding short-term transmission dynamics of methicillin-resistant *Staphylococcus aureus* in the patient room. *Infect Control Hosp Epidemiol* 2022;43(9):1147-1154.)

11. The text between 255-268 is hard to follow. This text is written in a way that makes it hard to get the point through the details.

Response: *We thank the reviewer for this comment. We have clarified each individual point below.*

11a. For example, here you say “Among the 12 transmissions with no microbiologically detected source during the interactive visit, WGS confirmed the presence of an identical strain found on the participant or in their room at an earlier visit 66.7% (8/12) of the time.” If the strain was present on that individual, how is this considered a transmission event? It's important to distinguish between translocation from one body site to another and transmission from one individual or environmental source to another.

Response: *Our definition of a transmission during an interactive visit simply put is: when a surface or hand is negative the first time we swab it and then positive the second time we swab it (following use or contact). Thus here, we describe 12 transmission events where either a surface or a participant's hand went from being negative to being positive for an MDRO, and it was not clear within that interactive visit alone where the MDRO came from (i.e., no microbiologically detected source). In these cases, we then looked at the previous in-room visits we had conducted with that participant and found that the identical organism was present on the participant or in their room most (67%) of the time. This allows us to conclude that the participant is most often the source of transmissions during interactive visits. The fact that we were not always seeing a clear source of transmission during an interactive visit suggests that either the participant's carriage of the organism was low density or quantity such that it could not be detected by our culture methods.*

11b. Similarly, “WGS confirmed identical strains between transmission source and destination surface 100% (11/11) of the time.” Would be better described as the number of instances of transmission from an environmental source to a specific body site vs. other patterns of transmission.

Response: *We have added text to the Results to further describe these 11 transmissions (lines 202-205). We added the same kind of information to described the 12 transmissions with an unknown source (lines 209-211).*

11c. Likewise, I'm getting lost in the numbers. There were 14 individuals for whom a transmission event was identified. 7 had a confirmed source of transmission. 9 had an unknown source of transmission. This is 16 individuals. This implies that 2 individuals had both a known and an unknown source of transmission. That's an important detail that gets lost in the current representation of the data. The authors are encouraged to present a visual figure of the text so that the text is not so arduous.

Response: *Lines 198-200 explain why the 9 and 7 = 16 does not equal 14—“ Two participants had instances of both unknown and known sources.” Details of these two participants are*

elaborated below so you see why we described in this way. We have also added all genomic data across all visits for these two participants to the Supplement (Supplemental Figure 5).

- **Participant 1044**
 - *Interactive visit #1 (11/9) to OT – VRE detected on participant’s hand at the start of the session. Transmission of VRE occurs on a toilet seat (i.e., toilet seat goes from MDRO-negative to VRE-positive following participant use)*
 - *Focusing on microbiology results from this visit alone, this is a transmission with a known source of transmission—VRE is transmitted from the participant’s hand to the toilet seat.*
 - *Interactive visit #2 (11/9) to Radiology for an x-ray – the participant’s hand is MDRO-negative at the start of the session, but becomes positive for VRE at the end of the session, after contact with one clean surface. No other VRE is detected during this visit.*
 - *Focusing on microbiology results from this visit alone, this is a transmission with an unknown source—VRE is transmitted to the participant hand from an unknown location.*
- **Participant 1064**
 - *Interactive visit #1 (3/3) to Radiology for an x-ray – MRSA detected on participant’s hand at the start of the session. Transmission of MRSA occurs on a bed/mat surface (i.e., bed/mat surface goes from MDRO-negative to MRSA-positive following participant use)*
 - *Focusing on microbiology results from this visit alone, this is a transmission with a known source of transmission—MRSA is transmitted from the participant’s hand to the bed/mat.*
 - *Interactive visit #2 (3/9) to PT – the participant’s hand is MDRO-negative at the start of the session, but becomes positive for MRSA at the end of the session, after contact with multiple clean surfaces. No other MRSA is detected during this visit.*
 - *Focusing on microbiology results from this visit alone, this is a transmission with an unknown source—MRSA is transmitted to the participant hand from an unknown location.*

11d. Line 266: 7 instances “among these participants.” Please specify how many participants. It would be preferable if the authors walked the reader through something like figure 6 illustrating the patterns discussed in this paragraph, rather than concluding with a pointer to a figure that the readers must make sense of on their own. This reviewer believes that representations like figure 6 should be more common in this manuscript so that the patterns of transmission and translocation can be deduced across subjects. This would make the data actionable in a way that summary statistics just don’t.

Response: *“Among these 14 participants,” has been clarified in the results (line 215). We agree that more representations like former Figure 6 (now Figure 4) are needed in this paper; we have added Supplemental Figure 5 with two more examples of participant stories over the course of their study period. We have added much detail to the Figure 4 legend (former Figure 6) as well to walk the reader through each participant visit.*

12. In the discussion, the authors mention, “We found that participant hands transmitted both MRSA and VRE, but not R-GNB, to environmental surfaces outside of their rooms.” This raises the unanswered question as to whether these species were co-transmitted with one another or whether these transmission events occurred independently among different individuals. Again, this highlights the need for the results to be presented on a per-subject manner.

Response: We meant to say MRSA or VRE, we apologize for the confusion. We have modified this part of the discussion (line 242) to clarify that participant hands are more likely to transmit MRSA during an interactive visit OR VRE during an interactive visit, but we saw few transmissions of RGNB during interactive visits. Data on the number of transmissions involving a single organism (MRSA only, VRE only) and more than one organism (MRSA and VRE) has been added to the results (lines 174-176). Three visits involved transmission of more than one organism (described below):

- Participant 1044, interactive visit 2: Participant hand negative at start of the interactive visit, but positive for both MRSA and VRE following contact with a negative table top.
- Participant 1087, interactive visit 1: Participant hand was positive for VRE at the start of the interactive visit. The participant used a clean (MDRO-negative) walker for 6 minutes, after which time the walker was contaminated with both MRSA and VRE.
- Participant 2059, interactive visit 1: Participant hand was positive for VRE at the start of the interactive visit. The participant touched an EKG keyboard and EKG wires/clips during the visit, both of which were MDRO-negative before contact, but after contact **both** the EKG keyboard and EKG wires/clips were positive for both MRSA and VRE.

13. The authors note that patterns were studied across 3 NHs but do not mention whether patterns of transmission differed across facilities in the main text. This seems like something worth at least a supplementary figure.

Response: Patterns of transmission across facilities are now described in Table 5 and the Results (lines 165-170).

14. “New MDRO acquisition in the study population was balanced by spontaneous loss of MDROs. Overall, 24 of 117 (20.5%) participants not colonized with an MDRO on enrollment later acquired and were discharged with a new MDRO.”

14a. The authors should perhaps break this down by MDRO, rather than presenting the data in the aggregate. Do we see differences in the rates of acquisition or loss by pathogen? Do we see differences in the number of people who are acquired by all pathogens vs. 1 pathogen who are discharged with or without a new MDRO

Response: Rates of new acquisition are now presented by MDRO and by facility in Table 4, and Figure 2 (former Figure 4) has been stratified by facility in the Supplement (Supplemental Figure 4).

Data reproducibility

14b. The data and code availability section is missing.

Response: We thank for reviewer for pointing this out. All Nature Communications formatting has now been followed, including the Data Availability section (lines 475-484). We did not include the optional ‘Code Availability’ section, as we employed no custom codes nor algorithms.

14c. SNP calling – the authors are encouraged to report the specific commands used to do their bioinformatics analysis. The current method section is insufficient for another investigator to reproduce the findings.

Response: We apologize for the lack of detail and have now included more detail in the methods (lines 383-388; 390-392; & 393-401).

14ci. “sequencing, of which 414 isolates from 26 patients passed QC”: please specify the QC protocol used and the method used to confirm species identification (16s gene? ANI?)

Response: We have now included details on our QC pipeline and species identification strategy (lines 372-373).

14cii. Why were the reference genomes for bacterial read mapping chosen? Had you chosen a strain from your outbreak would we have seen more core genome SNPs? How many reads mapped to these alignments and what fraction of the genome did the total alignment cover?

Response: We chose ST-specific reference genomes. As mentioned above, while including resident-specific reference genomes may have increased resolution, this would have complicated comparisons across residents. Moreover, as mentioned in our response to Reviewer 3, comment 5g, our analysis indicate that we have sufficient resolution to distinguish among circulating strains in the facility in the context of our study questions. We agree that maximizing resolution of genomic data is important in the context of comprehensive outbreak tracking among a densely sampled population, which we have employed in the past in the context of such study questions (Snitkin et al., Integrated genomic and interfacility patient-transfer data reveal the transmission pathways for multidrug-resistant *Klebsiella pneumoniae* in a regional outbreak. *Science Translational Medicine* 2017;9(417):eaan0093). The total percentage of the reference genomes under consideration for our analysis were 77% (2,199,367 / 2,872,915) for *E. faecium*, 83% (2,677,413 / 3,218,031) for *E. faecalis* and 87% (2,547,304 / 2,872,915) for *S. aureus* (lines 390-392).

14ciii. "variants were called and filtered using Samtools v.1.11" please specify the commands used to call and filter the SNPs.

Response: We have now added these details (lines 383-388).

14civ. Only variants located at nucleotide positions present in all isolates were considered for the transmission analyses: does this mean that positions with Ns were eliminated? What % of positions had Ns?

Response: Positions that were either not present in all isolates (i.e. non-core) or filtered in at least one isolate were removed from consideration due to not being robust markers of vertical descent, as mentioned above. The total number of considered positions are stated above (Reviewer 3, comment 14cii, lines 390-392).

14cv. How was mlst defined?

Response: We used the mlst tool, now cited in the manuscript (line 380, reference 35).

14d. All the tools used for bioinformatics analysis should be reported in the reporting summary along with their version numbers

Response: We have now added this information in our Reporting Summary.

14e. The reporting summary does not describe the full experimental design including the numbers of individuals at each of the three nursing homes

Response: Our reporting summary has been updated.

14f. "We followed enrolled participants for up to three months or until discharge, whichever came first." Please specify the interval of sample collection.

Response: The interval of sample collection for in-room and interactive visits has been clarified in the Methods (lines 309-315, see our response to Reviewer 3, Comments 2a, 3 and 7).

15. Discussion

15a. The higher proportion of VRE and R-GNB colonization at baseline and acquisition while in

the VA NH may be due to differences in the inherent transmissibility of these organisms or may reflect the long-term effects of the enterprise-wide MRSA surveillance policy implemented by the VA. <<< could this also be explained by the samples that you took? Would we expect to see different patterns if we sampled different body sites?

Response: *The reviewer brings up an excellent point. The sites that we sampled are typical sites for MRSA sampling. The reviewer might be interested in the following paper as well (Gontjes KJ, et al. Can alternative anatomical sites and environmental surveillance replace perianal screening for multidrug-resistant organisms in nursing homes? Infect Control Hosp Epidemiol 2022;43(8):1063-1066). Thus, with the participant body sites and room surfaces we sampled, we have confidence in our colonization and contamination rates detected for MRSA, VRE, and R-GNB.*

Minor points

1. Using newer genomic methods: standard genome sequencing isn't exactly new....I suggest just saying using genomics

Response: *We have made this change (line 59).*

REVIEWERS' COMMENTS

Reviewer #1 (Remarks to the Author):

The authors have thoroughly and expertly addressed all feedback that I provided to them. This paper provides noteworthy results based on robust methods that should be published.

Emily Sickbert-Bennett

Response: *We thank Ms. Sickbert-Bennet for her positive comments and recommendation for publication.*

Reviewer #2 (Remarks to the Author):

Thank you for the robust revisions and carefully addressing the comments. The added detail and explanations, in addition to the revised and novel figures, make this a clear and informative manuscript.

Response: *We thank Reviewer #2 for their positive comments and recommendation for publication.*

Reviewer #3 (Remarks to the Author):

The manuscript epidemiology of multidrug resistant organisms across three VA nursing homes and MDRO transmission during interactive visits by Mody et al. is much improved. The authors addressed the bulk of my concerns, and in its current form I believe the manuscript will be a valuable addition to the literature. I still take issue with the claim that they are observing spontaneous decolonization. It would be clearer to relabel 'spontaneously decolonized' as 'not detected' in figure 2 and elsewhere where that term is used. The term 'decolonized' implies a definitive outcome, but I'm concerned that what's described as 'spontaneously decolonized' could be a result of reaching the limit of detection (LOD) in addition to decolonization. To establish decolonization, a formal limit of detection analysis should be performed. This is the only way that a claim of decolonization can be rigorously supported. The unknown transmission event in participant 2, for example, could be an instance of this strain falling below the limit of detection on May 3, May 5 and May 10 on the hands. Throughout the manuscript, it is the opinion of this reviewer that the authors should soften language arguing that they are observing decolonization by using "not detected" if a LOD analysis is not performed.

Response: *We thank Reviewer #3 for their positive comments and recommendation for publication. We have made the requested change in Figure 2 and throughout the paper, replacing "spontaneously decolonized" with "not detected" or "no longer detected."*

Minor points:

1. Figure 1

- a. In the figure legend for figure 1, state the statistical test used
- b. Please state N for Facility A, B, C in the figure legend

Response: *We have added the statistical test used and the N for facilities A, B, and C in the Figure 1 legend.*

2. Supplementary figure 2

a. Please state how the combinations are plotted. I would've expected that Hands and Nares would represent a composite of both the hands and nares. Rather, it looks like maybe only people who are colonized by the same species at the hands and nares is shown. Please clarify in the figure legend.

Response: *We thank the reviewer for pointing this out. We have clarified in figure legend the combinations of participant body sites included on the y-axis of Supplemental Figure 2.*